# Maximum Entropy Reinforcement Learning with Diffusion Policy

**Xiaoyi Dong** [1 2]   **Jian Cheng** [1 3 4]   **Xi Sheryl Zhang** [1 4]

## Abstract

The Soft Actor-Critic (SAC) algorithm with a Gaussian policy has become a mainstream implementation for realizing the Maximum Entropy Reinforcement Learning (MaxEnt RL) objective, which incorporates entropy maximization to encourage exploration and enhance policy robustness. While the Gaussian policy performs well on simpler tasks, its exploration capacity and potential performance in complex multi-goal RL environments are limited by its inherent unimodality. In this paper, we employ the diffusion model, a powerful generative model capable of capturing complex multimodal distributions, as the policy representation to fulfill the MaxEnt RL objective, developing a method named *MaxEnt RL with Diffusion Policy* (MaxEntDP). Our method enables efficient exploration and brings the policy closer to the optimal MaxEnt policy. Experimental results on Mujoco benchmarks show that MaxEntDP outperforms the Gaussian policy and other generative models within the MaxEnt RL framework, and performs comparably to other state-of-the-art diffusion-based online RL algorithms. Our code is available at https://github.com/diffusionyes/MaxEntDP.

## 1. Introduction

Reinforcement Learning (RL) has emerged as a powerful paradigm for training intelligent agents to make decisions in complex control tasks (Silver et al., 2016; Mnih et al., 2015; Kaufmann et al., 2023; Kiran et al., 2021; Ibarz et al., 2021). Traditionally, RL focuses on maximizing the expected cumulative reward, where the agent selects actions that yield the highest return in each state (Sutton & Barto, 1999). However, this approach often overlooks the inherent uncertainty and variability of real-world environments, which can lead to suboptimal or overly deterministic policies. To address these limitations, Maximum Entropy Reinforcement Learning (MaxEnt RL) incorporates entropy maximization into the standard RL objective, encouraging exploration and improving robustness during policy learning (Toussaint, 2009; Ziebart, 2010; Haarnoja et al., 2017).

The Soft Actor-Critic (SAC) algorithm (Haarnoja et al., 2018) is an effective method for achieving the MaxEnt RL objective, which alternates between policy evaluation and policy improvement to progressively refine the policy. With high-capacity neural network approximators and suitable optimization techniques, SAC can provably converge to the optimal MaxEnt policy within the chosen policy set. The choice of policy representation in SAC is crucial, as it influences the exploration behavior during training and determines the proximity of the candidate policies to the optimal MaxEnt policy. In complex multi-goal RL tasks, where multiple feasible behavioral modes exist, the commonly used Gaussian policy typically explores only a single mode, which can cause the agent to get trapped in a local optimum and fail to approach the optimal MaxEnt policy that captures all possible behavioral modes.

In this paper, we propose using diffusion models (Sohl-Dickstein et al., 2015; Song & Ermon, 2019; Ho et al., 2020; Song et al., 2021b), a powerful generative model, as the policy representation within the SAC framework. This allows for the exploration of all promising behavioral modes and facilitates convergence to the optimal MaxEnt policy. Diffusion models transform the original data distribution into a tractable Gaussian by progressively adding Gaussian noise, which is known as the forward diffusion process. After training a neural network to predict the noise added to the noisy samples, the original data can be recovered by solving the reverse diffusion process with the noise prediction network. While several generative models, e.g., variational autoencoders (Kingma, 2013), generative adversarial networks (Goodfellow et al., 2020), and normalizing flows (Rezende & Mohamed, 2015) could serve as the policy representation, we choose diffusion models due to their balance between expressiveness and inference speed, achieving remarkable performance with affordable training and inference costs.

[1]C²DL, Institute of Automation, Chinese Academy of Sciences [2]School of Artificial Intelligence, University of Chinese Academy of Sciences [3]School of Future Technology, University of Chinese Academy of Sciences [4]AiRiA. Correspondence to: Xi Sheryl Zhang <sheryl.zhangxi@gmail.com>.

*Proceedings of the 42nd International Conference on Machine Learning*, Vancouver, Canada. PMLR 267, 2025. Copyright 2025 by the author(s).

However, integrating diffusion models into the SAC framework presents two key challenges: 1) How to train a diffusion model to approximate the exponential of the Q-function in the policy improvement step? 2) How to compute the log probability of the diffusion policy when evaluating the soft Q-function? To address the first challenge, we analyze the training target of the noise prediction network in diffusion models and propose a Q-weighted Noise Estimation method. For the second challenge, we introduce a numerical integration technique to approximate the log probability of the diffusion model. We evaluate the effectiveness of our approach on Mujoco benchmarks. The experimental results demonstrate that our method outperforms the Gaussian policy and other generative models within the MaxEnt RL framework, and performs comparably to other state-of-the-art diffusion-based online RL algorithms.

## 2. Preliminary

### 2.1. Maximum Entropy Reinforcement Learning

In this paper, we focus on policy learning in continuous action spaces. We consider a Markov Decision Process (MDP) defined by the tuple $(\mathcal{S}, \mathcal{A}, p, r, \rho_0, \gamma)$, where $\mathcal{S}$ represents the state space, $\mathcal{A}$ is the continuous action space, $p : \mathcal{S} \times \mathcal{S} \times \mathcal{A} \to [0, +\infty]$ is the probability density function of the next state $\boldsymbol{s}_{t+1} \in \mathcal{S}$ given the current state $\boldsymbol{s}_t \in \mathcal{S}$ and the action $\boldsymbol{a}_t \in \mathcal{A}$, $r : \mathcal{S} \times \mathcal{A} \to [r_{\min}, r_{\max}]$ is the bounded reward function, $\rho_0 : \mathcal{S} \to [0, +\infty]$ is the distribution of the initial state $\boldsymbol{s}_0$ and $\gamma \in [0, 1]$ is the discount factor. The marginals of the trajectory distribution induced by a policy $\pi(\boldsymbol{a}_t|\boldsymbol{s}_t)$ are denoted as $\rho_\pi(\boldsymbol{s}_t, \boldsymbol{a}_t)$.

The standard RL aims to learn a policy that maximizes the expected cumulative reward. To encourage stochastic policies, Maximum Entropy RL augments this objective by incorporating the expected entropy of the policy:

$$\mathrm{J}(\pi) = \sum_{t=0}^{\infty} \gamma^t \mathbb{E}_{(\boldsymbol{s}_t, \boldsymbol{a}_t) \sim \rho_\pi} \left[ r(\boldsymbol{s}_t, \boldsymbol{a}_t) + \beta \mathcal{H}(\pi(\cdot|\boldsymbol{s}_t)) \right], \quad (1)$$

where $\mathcal{H}(\pi(\cdot|\boldsymbol{s}_t)) = \mathbb{E}_{\boldsymbol{a}_t \sim \pi(\cdot|\boldsymbol{s}_t)} \left[ -\log \pi(\boldsymbol{a}_t|\boldsymbol{s}_t) \right]$, and $\beta$ is the temperature parameter that controls the trade-off between the entropy and reward terms. A higher value of $\beta$ drives the optimal policy to be more stochastic, which is advantageous for RL tasks requiring extensive exploration. In contrast, the standard RL objective can be seen as the limiting case where $\beta \to 0$.

### 2.2. Soft Actor Critic

The optimal maximum entropy policy can be derived by applying the Soft Actor-Critic (SAC) algorithm (Haarnoja et al., 2018). In this subsection, we will briefly introduce the framework of SAC, and the relevant proofs are provided in Appendix A.1. The SAC algorithm utilizes two parameterized networks, $Q_\theta$ and $\pi_\phi$, to model the soft Q-function

and the policy, where $\theta$ and $\phi$ represent the parameters of the respective networks. These networks are optimized by alternating between policy evaluation and policy improvement.

In the policy evaluation step, the soft Q-function of the current policy $\pi_\phi$ is learned by minimizing the soft Bellman error:

$$L(\theta) = \mathbb{E}_{(\boldsymbol{s}, \boldsymbol{a}) \sim \mathcal{D}} \left[ \frac{1}{2} \left( Q_\theta(\boldsymbol{s}, \boldsymbol{a}) - \hat{Q}(\boldsymbol{s}, \boldsymbol{a}) \right)^2 \right], \quad (2)$$

where $\mathcal{D}$ is the replay buffer, and the target value $\hat{Q}(\boldsymbol{s}, \boldsymbol{a}) = r(\boldsymbol{s}, \boldsymbol{a}) + \gamma \mathbb{E}_{\boldsymbol{s}' \sim p, \boldsymbol{a}' \sim \pi_\phi} \left[ Q_\theta(\boldsymbol{s}', \boldsymbol{a}') - \beta \log \pi_\phi(\boldsymbol{a}'|\boldsymbol{s}') \right]$.

In the policy improvement step, the old policy $\pi_{\phi_k}$ is updated towards the exponential of the new Q-function, whose soft value is guaranteed higher than the old policy. However, the target policy may be too complex to be exactly represented by any policy within the parameterized policy set $\Pi = \{\pi_\phi | \phi \in \Phi\}$, where $\Phi$ is the parameter space of the policy. Therefore, the new policy is obtained by projecting the target policy onto the policy set $\Pi$ based on the Kullback-Leibler divergence:

$$L(\phi) = \mathrm{D}_{\mathrm{KL}} \left( \pi_\phi(\cdot|\boldsymbol{s}) \left\| \frac{\exp(\frac{1}{\beta} Q_\theta(\boldsymbol{s}, \cdot))}{Z_\theta(\boldsymbol{s})} \right. \right). \quad (3)$$

**Theorem 2.1.** *(Soft Policy Iteration) In the tabular setting, let $L(\theta_k) = 0$ and $L(\phi_k)$ be minimized for each $k$. Repeated application of policy evaluation and policy improvement, i.e., $k \to \infty$, $\pi_{\phi_k}$ will converge to a policy $\pi^*$ such that $Q^{\pi^*}(\boldsymbol{s}, \boldsymbol{a}) \geq Q^\pi(\boldsymbol{s}, \boldsymbol{a})$ for all $\pi \in \Pi$ and $(\boldsymbol{s}, \boldsymbol{a}) \in \mathcal{S} \times \mathcal{A}$ with $|\mathcal{A}| < \infty$.*

Theorem 2.1 suggests that if the Bellman error can be reduced to zero and the policy loss is minimized at each optimization step, the soft actor-critic algorithm will converge to the optimal maximum entropy policy within the policy set $\Pi$. This indicates that the choice of the policy set $\Pi$ significantly affects the performance of the soft actor-critic algorithm. Specifically, a more expressive policy class will yield a policy closer to the optimal MaxEnt policy. Inspired by this intuition, we employ the diffusion model to represent the policy, as it is highly expressive and well-suited to capture the complex multimodal distribution (Chi et al., 2023; Wang et al., 2023; Chen et al., 2023; Ajay et al., 2023).

### 2.3. Diffusion Models

Diffusion models are powerful generative models. Given an unknown data distribution $p(\boldsymbol{x}_0)$, which is typically a mixture of Dirac delta measures over the training dataset, diffusion models transform this data distribution into a tractable Gaussian distribution by progressively adding Gaussian noise (Ho et al., 2020). In the context of a Variance-Preserving (VP) diffusion process (Ho et al., 2020; Song

et al., 2021b), the transition from the original sample $\boldsymbol{x}_0$ at time $t = 0$ to the noisy sample $\boldsymbol{x}_t$ at time $t \in [0, 1]$ follows the distribution:

$$p(\boldsymbol{x}_t | \boldsymbol{x}_0) = \mathcal{N}(\boldsymbol{x}_t | \sqrt{\sigma(\alpha_t)} \boldsymbol{x}_0, \sigma(-\alpha_t) \boldsymbol{I}), \quad (4)$$

where $\alpha_t$ represents the log of the Signal-to-Noise Ratio (SNR) at time $t$, and $\sigma(\cdot)$ is the sigmoid function. $\alpha_t$ determines the amount of noise added at each time and is referred to as the noise schedule of a diffusion model. Denote the marginal distribution of $\boldsymbol{x}_t$ as $p(\boldsymbol{x}_t)$. The noise schedule should be designed to ensure that $p(\boldsymbol{x}_1 | \boldsymbol{x}_0) \approx p(\boldsymbol{x}_1) \approx \mathcal{N}(\boldsymbol{x}_1 | \boldsymbol{0}, \boldsymbol{I})$, and that $\alpha_t$ is strictly decreasing w.r.t. $t$. Then, starting from $\boldsymbol{x}_1 \sim \mathcal{N}(\boldsymbol{x}_1 | \boldsymbol{0}, \boldsymbol{I})$, the original data samples can be recovered by reversing the diffusion process from $t = 1$ to $t = 0$. For sample generation, we can also employ the following probability flow ordinary differential equation (ODE) that shares the same marginal distribution with the diffusion process (Song et al., 2021b):

$$\frac{\mathrm{d}\boldsymbol{x}_t}{\mathrm{d}t} = f(t)\boldsymbol{x}_t - \frac{1}{2} g^2(t) \nabla_{\boldsymbol{x}_t} \log p(\boldsymbol{x}_t), \quad (5)$$

where $f(t) = \frac{1}{2} \frac{\mathrm{d} \log \sigma(\alpha_t)}{\mathrm{d}t}$, $g^2(t) = -\frac{\mathrm{d} \log \sigma(\alpha_t)}{\mathrm{d}t}$, and $\nabla_{\boldsymbol{x}_t} \log p(\boldsymbol{x}_t)$, known as the score function, is the only unknown term. Consequently, diffusion models train a neural network $\boldsymbol{\epsilon}_\phi(\boldsymbol{x}_t, \alpha_t)$ to approximate the scaled score function $-\sqrt{\sigma(-\alpha_t)} \nabla_{\boldsymbol{x}_t} \log p(\boldsymbol{x}_t)$. The training loss $L(\phi)$ is defined as:

$$L(\phi) = \mathbb{E}_{t,\boldsymbol{x}_t} \left[ w_t \left\| \boldsymbol{\epsilon}_\phi(\boldsymbol{x}_t, \alpha_t) + \sqrt{\sigma(-\alpha_t)} \nabla_{\boldsymbol{x}_t} \log p(\boldsymbol{x}_t) \right\|_2^2 \right] \tag{6}$$

$$= \mathbb{E}_{t,\boldsymbol{x}_0,\boldsymbol{\epsilon}} \left[ w_t \left\| \boldsymbol{\epsilon}_\phi(\boldsymbol{x}_t, \alpha_t) - \boldsymbol{\epsilon} \right\|_2^2 \right] + C \tag{7}$$

where $\boldsymbol{x}_0 \sim p(\boldsymbol{x}_0)$, $\boldsymbol{\epsilon} \sim \mathcal{N}(\boldsymbol{0}, \boldsymbol{I})$, $t \sim \mathcal{U}([0, 1])$, $\boldsymbol{x}_t = \sqrt{\sigma(\alpha_t)} \boldsymbol{x}_0 + \sqrt{\sigma(-\alpha_t)} \boldsymbol{\epsilon}$, $w_t$ is a weighting function and usually set to $w_t \equiv 1$, and $C$ is a constant independent of $\phi$. In this setup, the network $\boldsymbol{\epsilon}_\phi(\boldsymbol{x}_t, \alpha_t)$ target at predicting the expectation of noise added to the noisy sample $\boldsymbol{x}_t$, and is therefore called the noise prediction network. Minimizing the loss function $L(\phi)$ results in the following relationship:

$$\nabla_{\boldsymbol{x}_t} \log p(\boldsymbol{x}_t) = -\frac{\boldsymbol{\epsilon}_\phi(\boldsymbol{x}_t, \alpha_t)}{\sqrt{\sigma(-\alpha_t)}}. \quad (8)$$

Then we can solve the probability flow ODE in Equation 5 with the assistance of existing ODE solvers (Ho et al., 2020; Song et al., 2021a; Lu et al., 2022; Karras et al., 2022; Zheng et al., 2023) to generate data samples.

## 3. Methodology

In the soft actor-critic algorithms, Gaussian policies have become the most widely used class of policy representation due to their simplicity and efficiency. Although Gaussian policies perform well in relatively simple single-goal RL environments, they often struggle with more complex multi-goal tasks.

Consider a typical RL task that involves multiple behavior modes. The most efficient solution is to explore all behavior modes until one obviously outperforms the others. However, this exploration strategy is difficult to achieve with Gaussian policies. In the training process of a soft actor-critic algorithm with Gaussian policies, minimizing the KL divergence between the Gaussian policy and the exponential of the Q-function—which is often multimodal in multi-goal tasks—tends to push the Gaussian policy to allocate most of the probability mass to the action region with the highest Q value (Chen et al., 2024a). Consequently, other promising action regions with slightly lower Q values will be neglected, which may cause the agent to become stuck at a local optimal policy.

However, an efficient exploration strategy can be achieved by replacing the Gaussian policy with a more expressive policy representation class. If accurately fitting the multimodal target policy (i.e., the exponential of the Q-function), the agent will explore all high-return action regions at a high probability, thus reducing the risk of converging to a local optimum. Moreover, recall that when the assumptions on loss optimization are met, the soft actor-critic algorithm is guaranteed to converge to the optimal maximum entropy policy within the chosen policy class. Therefore, with sufficient network capacity and appropriate optimization techniques, we can obtain the true optimal maximum entropy policy, as long as the selected policy representation class is expressive enough to capture it.

The above analysis emphasizes the importance of applying an expressive policy class to achieve efficient exploration as well as a higher performance upper bound. Since diffusion models have demonstrated remarkable performance in capturing complex multimodal distributions, we adopt them to represent the policy within the soft actor-critic framework. However, integrating a diffusion-based policy into the soft actor-critic algorithm presents several challenges: (1) In the policy improvement step, the new diffusion policy is updated to approximate the exponential of the Q-function. However, existing methods for training diffusion models rely on samples from the target distribution, which are unavailable in this case. (2) In the policy evaluation step, computing the soft Q-function requires access to the probability of the diffusion policy. Nevertheless, diffusion models implicitly model data distributions by estimating their score functions, making it intractable to compute the exact probability.

The remainder of this section addresses these challenges and describes how to incorporate diffusion models into the soft actor-critic algorithm for efficient policy learning. We first

propose the Q-weighted Noise Estimation approach to fit the exponential of the Q-function in Section 3.1, then introduce a method for probability approximation in diffusion policies in Section 3.2, and finally present the complete algorithm in Section 3.3. We name this method MaxEntDP because it can fulfill the MaxEnt RL objective with diffusion policies.

## 3.1. Q-weighted Noise Estimation

Given a Q-function $Q(s, a)$, below we will analyze how to train a noise prediction network $\epsilon_\phi$ in the diffusion model to approximate the target distribution:

$$\pi(a|s) = \frac{\exp(\frac{1}{\beta}Q(s, a))}{Z(s)}. \qquad (9)$$

Omitting the state in the condition for simplicity and following the symbol convention of diffusion models, we rewrite $\pi(a|s)$ as $p(a_0)$. The transition from the original action samples $a_0$ at time $t = 0$ to the noisy actions $a_t$ at time $t \in [0, 1]$ is defined as:

$$p(a_t|a_0) = \mathcal{N}(a_t|\sqrt{\sigma(\alpha_t)}a_0, \sigma(-\alpha_t)I) \qquad (10)$$

Note that the symbol $t$ stands for the time of diffusion models if not specified.

The marginal distribution of noisy actions $a_t$ at time $t$ is denoted by $p(a_t)$. To sample from $p(a_0)$, we need to estimate the score function $\nabla_{a_t} \log p(a_t)$ at each intermediate time $t$ during the diffusion process. The score function can be reformulated as:

$$\nabla_{a_t} \log p(a_t) = \mathbb{E}_{p(a_0|a_t)} \left[ \nabla_{a_t} \log p(a_t|a_0) \right], \qquad (11)$$

which is an expectation with respect to the conditional distribution $p(a_0|a_t)$, a.k.a. the reverse transition distribution of the diffusion process. If samples from $p(a_0)$ are available, as is often the case in the application scenarios of diffusion models (Saharia et al., 2022; Ho et al., 2022; Chi et al., 2023; Xu et al., 2023; Huang et al., 2023), we can first sample original actions $a_0 \sim p(a_0)$, and then sample noisy actions $a_t \sim p(a_t|a_0)$ to obtain several sample pairs following the joint distribution $p(a_0, a_t)$. Then for a fixed noisy action $a_t$, the corresponding $a_0$ will conform the conditional distribution $p(a_0|a_t)$, which can serve as Monte Carlo samples to estimate the expectation in Equation 11. Conversely, in the context of the soft actor-critic algorithm, we lack samples from the target distribution $p(a_0)$ but instead have access to a Q-function. Therefore, we must establish the relationship between the conditional distribution $p(a_0|a_t)$ and the Q-function.

**Lemma 3.1.** *(Decomposition of the Reverse Transition Distribution) The conditional distribution $p(a_0|a_t)$ can be decomposed as*

$$p(a_0|a_t) \propto \exp(\tfrac{1}{\beta}Q(a_0))\mathcal{N}(a_0|\tfrac{1}{\sqrt{\sigma(\alpha_t)}}a_t, \tfrac{\sigma(-\alpha_t)}{\sigma(\alpha_t)}I) \quad (12)$$

The proof is provided in Appendix A.2. Lemma 3.1 demonstrates that the conditional distribution $p(a_0|a_t)$ can be seen as a Gaussian distribution of $a_0$ weighted by the exponential of the Q-function. Sampling from the Gaussian distribution is straightforward, we can apply importance sampling (Bishop, 2006) to estimate the expectation in Equation 11.

**Theorem 3.2.** *(Importance Sampling Estimate for the Score Function) The score function can be estimated by*

$$\nabla_{a_t} \log p(a_t) \approx \frac{1}{\sqrt{\sigma(-\alpha_t)}} \cdot \frac{1}{K} \sum_{i=1}^{K} w(a_0^i)\epsilon^i, \qquad (13)$$

*where $\epsilon^1, \ldots, \epsilon^K \sim \mathcal{N}(0, I)$, $a_0^i = \frac{1}{\sqrt{\sigma(\alpha_t)}}a_t + \frac{\sqrt{\sigma(-\alpha_t)}}{\sqrt{\sigma(\alpha_t)}}\epsilon^i$ and the importance ratio $w(a_0) = \frac{\exp(\frac{1}{\beta}Q(a_0))}{Z(a_t)}$ with $Z(a_t)$ being the normalizing constant of $p(a_0|a_t)$.*

The derivation is detailed in Appendix A.3. Although this importance sampling estimate is unbiased, it exhibits high variance when the variance of the Q-function is large. To address this issue, we employ the weighted importance sampling approach (Bishop, 2006) to reduce variance and stabilize the training process.

**Theorem 3.3.** *(Weighted Importance Sampling Estimate for the Score Function) The score function can be estimated by*

$$\nabla_{a_t} \log p(a_t) \approx \frac{1}{\sqrt{\sigma(-\alpha_t)}} \cdot \sum_{i=1}^{K} \frac{w(a_0^i)}{\sum_{j=1}^{K} w(a_0^j)}\epsilon^i \qquad (14)$$

$$= \frac{1}{\sqrt{\sigma(-\alpha_t)}} \sum_{i=1}^{K} \mathrm{softmax}(\tfrac{1}{\beta}Q(a_0^{1:K}))_i\epsilon^i, \quad (15)$$

*where $\mathrm{softmax}(\tfrac{1}{\beta}Q(a_0^{1:K}))_i = \frac{\exp(\frac{1}{\beta}Q(a_0^i))}{\sum_{j=1}^{K} \exp(\frac{1}{\beta}Q(a_0^j))}$.*

The normalizing constant $Z(a_t)$ is canceled out in Equation 15, eliminating the need for its explicit computation. Since the bias of the weighted importance sampling method decreases as the number of Monte Carlo samples increases, a larger value of $K$ is preferred in practice given adequate computation budgets.

Then the training target of the noise prediction network is

$$\epsilon^*(a_t, \alpha_t) = -\sqrt{\sigma(-\alpha_t)}\nabla_{a_t} \log p(a_t) \qquad (16)$$

$$\approx -\sum_{i=1}^{K} \mathrm{softmax}(\tfrac{1}{\beta}Q(a_0^{1:K}))_i\epsilon^i, \qquad (17)$$

This target can be interpreted as a weighted sum of noise, with the weights being the exponential of the Q-value. Consequently, we refer to this method as Q-weighted Noise Estimation for training the noise prediction network. The overall training loss is

$$L(\phi) = \mathbb{E}_{p(a_t)} \left[ \| \epsilon_\phi(a_t, \alpha_t) - \epsilon^*(a_t, \alpha_t) \|_2^2 \right] \qquad (18)$$

While the true distribution of noisy actions $p(\boldsymbol{a}_t)$ may be inaccessible, we can substitute it with other distributions with full support, as the loss will still be minimized for each $\boldsymbol{a}_t$ given sufficient network capacity.

We briefly compare our method with two previous approaches that approximate the exponential of a given function $Q(a)$. The QSM method (Psenka et al., 2024) estimates the score function as $\nabla_{\boldsymbol{a}_t} \log p(\boldsymbol{a}_t) \approx \nabla_{\boldsymbol{a}_t} \frac{1}{\beta} Q(\boldsymbol{a}_t)$. This approximation requires $p(\boldsymbol{a}_t) \propto \exp(\frac{1}{\beta} Q(\boldsymbol{a}_t))$, which is true only when the time $t$ is close to $0$. Therefore, the score function estimation in QSM is imprecise for most values of $t$. Another method iDEM (Akhound-Sadegh et al., 2024) proposes $\nabla_{\boldsymbol{a}_t} \log p(\boldsymbol{a}_t) \approx \frac{1}{\sqrt{\sigma(\alpha_t)}} \sum_{i=1}^{K} \text{softmax}(\frac{1}{\beta} Q(\boldsymbol{a}_0^{1:K}))_i \nabla_{\boldsymbol{a}_0^i} \frac{1}{\beta} Q(\boldsymbol{a}_0^i)$, and the derivation is included in Appendix A.4 for completion. Although the expressions of iDEM and our method appear similar and both can approach the true score function as $K \to \infty$, our method does not require computing the gradient of the Q-function, which is more computationally efficient, especially when the Q-function is evaluated on a neural network. Furthermore, the experiments in Section 5.2 demonstrate that the variance of the score estimation in our method is significantly lower than the other two methods that rely on gradient computation, leading to a more stable training process.

### 3.2. Probability Approximation of Diffusion Policy

Diffusion models approximate the desired distributions by estimating their score function. Although this implicit modeling enhances the expressiveness of the model, enabling it to approximate any distribution with a differentiable probability density function, it also introduces challenges in computing the exact likelihood of the distribution.

Previous study (Kong et al., 2023; Wu et al., 2024) proved that the log-likelihood of $p(\boldsymbol{a}_0)$ can be written exactly as an expression that depends only on the true noise prediction target, i.e.,

$$\log p(\boldsymbol{a}_0) = c - \frac{1}{2} \int_{-\infty}^{+\infty} \mathbb{E}_{\boldsymbol{\epsilon}} \left[ \| \boldsymbol{\epsilon} - \boldsymbol{\epsilon}^*(\boldsymbol{a}_t, \alpha_t) \|_2^2 \right] \mathrm{d}\alpha_t \quad (19)$$

where $c = -\frac{d}{2} \log(2\pi e) + \frac{d}{2} \int_{-\infty}^{+\infty} \sigma(\alpha_t) \mathrm{d}\alpha_t$ with $d$ being the dimension of $\boldsymbol{a}_0$, $\boldsymbol{\epsilon} \sim \mathcal{N}(\boldsymbol{0}, \boldsymbol{I})$, $\boldsymbol{a}_t = \sqrt{\sigma(\alpha_t)}\boldsymbol{a}_0 + \sqrt{\sigma(-\alpha_t)}\boldsymbol{\epsilon}$, and $\boldsymbol{\epsilon}^*(\boldsymbol{a}_t, \alpha_t) = -\sqrt{\sigma(-\alpha_t)}\nabla_{\boldsymbol{a}_t} \log p(\boldsymbol{a}_t)$ is the training target of the noise prediction network.

**Corollary 3.4.** *(The Exact Probability of Diffusion Policy)* *Let $\boldsymbol{\epsilon}_\phi$ be a well-trained noise prediction network, i.e., it can induce a probability density function $p_\phi(\boldsymbol{a}_0)$ satisfying $\boldsymbol{\epsilon}_\phi(\boldsymbol{a}_t, \alpha_t) = -\sqrt{\sigma(-\alpha_t)}\nabla_{\boldsymbol{a}_t} \log p_\phi(\boldsymbol{a}_t)$, then*

$$\log p_\phi(\boldsymbol{a}_0) = c - \frac{1}{2} \int_{-\infty}^{+\infty} \mathbb{E}_{\boldsymbol{\epsilon}} \left[ \| \boldsymbol{\epsilon} - \boldsymbol{\epsilon}_\phi(\boldsymbol{a}_t, \alpha_t) \|_2^2 \right] d\alpha_t \quad (20)$$

This corollary can be inferred from Equation 19. However,

this expression is intractable because both the integral in $c$ and the integral of the noise prediction error diverge, with only their difference converging (Kong et al., 2023). We attempt to approximate the integral using numerical integration techniques. However, we observe that using the log SNR as the integration variable results in a high variance, as it spans from $-\infty$ to $+\infty$. Therefore, we instead utilize $\sigma(\alpha_t)$ with a narrower integration domain of $(0, 1)$.

**Theorem 3.5.** *(The Probability Approximation of Diffusion Policy)* *The log probability of diffusion policy can be approximated by*

$$\log p_\phi(\boldsymbol{a}_0) \approx c' + \frac{1}{2} \sum_{i=1}^{T} w_{t_i} \left( d \cdot \sigma(\alpha_{t_i}) - \tilde{\boldsymbol{\epsilon}}_\phi(\boldsymbol{a}_{t_i}, \alpha_{t_i}) \right) \quad (21)$$

*where $c' = -\frac{d}{2} \log(2\pi e)$, $t_{0:T}$ are uniformly spaced timesteps in $[t_{min}, t_{max}]$, $w_{t_i} = \frac{\sigma(\alpha_{t_{i-1}}) - \sigma(\alpha_{t_i})}{\sigma(\alpha_{t_i})\sigma(-\alpha_{t_i})}$ is the weight at $t_i$, $\tilde{\boldsymbol{\epsilon}}_\phi(\boldsymbol{a}_{t_i}, \alpha_{t_i}) = \frac{1}{N} \sum_{j=1}^{N} \| \boldsymbol{\epsilon}^j - \boldsymbol{\epsilon}_\phi(\boldsymbol{a}_{t_i}^j, \alpha_{t_i}) \|_2^2$ is the noise prediction error estimation at $t_i$.*

The detailed derivation is provided in Appendix A.5.

### 3.3. MaxEnt RL with Diffusion Policy

After addressing the critical challenges in training and probability estimation for the diffusion policy, we present the complete algorithm for achieving the MaxEnt RL objective with a diffusion policy. Our approach is based on the soft actor-critic framework. We utilize two neural networks: $Q_\theta(s, a)$ to model the Q-function, and $\boldsymbol{\epsilon}_\phi(\boldsymbol{a}_t, \alpha_t, \boldsymbol{s})$ to model the noise prediction network for the diffusion policy $\pi_\phi(\boldsymbol{a}_0|\boldsymbol{s})$.

The training process alternates between policy evaluation and policy improvement. In the policy evaluation step, the Q-network is trained by minimizing the soft Bellman error, as defined in Equation 2. Here, the actions $a' \sim \pi_\phi(\cdot|\boldsymbol{s'})$ are sampled by solving the probability flow ODE in Equation 5 with the noise prediction network $\boldsymbol{\epsilon}_\phi(\boldsymbol{a}_t, \alpha_t, \boldsymbol{s})$, and the log probality $\log \pi_\phi(\cdot|\boldsymbol{s})$ is approximated using Equation 21. In the policy improvement step, the noise prediction network is optimized using the loss function in Equation 18[1], with the training target computed in Equation 17. The pseudocode for our method is presented in Algorithm 1.

In addition, we adopt several techniques to improve the training and inference of our method:

**Truncated Gaussian Noise Distribution for Bounded Action Space.** In RL tasks with bounded action spaces, the Q-function is undefined outside the action space. To avoid evaluating Q-values for illegal actions, the noise distribution in Equation 17 is modified from a standard Gaussian to a

---

[1]The minimizers of Equation 18 and 3 will be equal when the exponential of the Q-function can be exactly expressed by the chosen policy set, so the capacity of the noise prediction network is preferred to be large if allowed.

**Algorithm 1** MaxEnt RL with Diffusion Policy

1: Initialize critic networks $Q_{\theta_1}$, $Q_{\theta_2}$, and the noise prediction network $\boldsymbol{\epsilon}_\phi$ with random parameters $\theta_1, \theta_2, \phi$.
2: Initialize target networks $\theta_1' \leftarrow \theta_1, \theta_2' \leftarrow \theta_2$
3: Initialize replay buffer $\mathcal{D}$
4: **for** each iteration **do**
5:      **for** each sampling step **do**
6:          Sample $\boldsymbol{a} \sim \pi_\phi(\cdot|\boldsymbol{s})$ according to Equation 5
7:          Step environment: $\boldsymbol{s}', r \leftarrow \text{env}(\boldsymbol{a})$
8:          Store $(\boldsymbol{s}, \boldsymbol{a}, r, \boldsymbol{s}')$ in $\mathcal{D}$
9:      **end for**
10:      **for** each update step **do**
11:          Sample $B$ transitions $(\boldsymbol{s}, \boldsymbol{a}, r, \boldsymbol{s}')$ from $\mathcal{D}$
12:          Sample $\boldsymbol{a}' \sim \pi_\phi(\cdot|\boldsymbol{s}')$ according to Equation 5
13:          Compute $\log \pi_\phi(\boldsymbol{a}'|\boldsymbol{s}')$ using Equation 21
14:          Compute the target Q-value: $\hat{Q}(\boldsymbol{s}, \boldsymbol{a}) = r(\boldsymbol{s}, \boldsymbol{a}) + \gamma \left(\min_{i=1,2} Q_{\theta_i}(\boldsymbol{s}', \boldsymbol{a}') - \beta \log \pi_\phi(\boldsymbol{a}'|\boldsymbol{s}')\right)$.
15:          Update critics: $\theta_i = \arg\min_{\theta_i} \frac{1}{B} \sum (Q_{\theta_i}(\boldsymbol{s}, \boldsymbol{a}) - \hat{Q}(\boldsymbol{s}, \boldsymbol{a}))^2$
16:          Sample $t \sim \mathcal{U}([t_{\min}, t_{\max}])$ and the noisy action $\boldsymbol{a}_t \sim \mathcal{N}(\boldsymbol{a}_t|\sqrt{\sigma(\alpha_t)}\boldsymbol{a}, \sigma(-\alpha_t)\boldsymbol{I})$
17:          Estimate $\boldsymbol{\epsilon}^*(\boldsymbol{a}_t, \alpha_t, \boldsymbol{s})$ with Equation 17
18:          Update the noise prediction network: $\phi = \arg\min_\phi \frac{1}{B} \sum \| \boldsymbol{\epsilon}_\phi(\boldsymbol{a}_t, \alpha_t, \boldsymbol{s}) - \boldsymbol{\epsilon}^*(\boldsymbol{a}_t, \alpha_t, \boldsymbol{s}) \|_2^2$
19:          Updtae target networks: $\theta_i \leftarrow \tau\theta_i + (1-\tau)\theta_i$
20:      **end for**
21: **end for**

truncated standard Gaussian. This modification still generates samples according to the Gaussian function, but all samples are bounded in the specified range.

**Action Selection for Inference.** Previous studies (Chao et al., 2024; Wang et al., 2023; Mao et al., 2024; Chen et al., 2024b) have found that a deterministic policy typically outperforms its stochastic counterpart during testing. Consequently, we employ an action selection technique to further refine the policy after training. Specifically, $M$ action candidates are sampled from the diffusion policy, and the action with the highest Q-value is selected to interact with the RL environment.

## 4. Related Work

**MaxEnt RL** A variety of approaches have been proposed to achieve the MaxEnt RL objective. SQL (Haarnoja et al., 2017) introduces soft Q-learning to learn the optimal soft Q-function and trains an energy-based model using the amortized Stein variational gradient method to generate actions according to the exponential of the optimal soft Q-function. SAC (Haarnoja et al., 2018) presents the soft actor-critic algorithm, which iteratively improves the policy towards a higher soft value, and provides an implementation using

Gaussian policies. To improve the sample efficiency of SAC, CrossQ (Bhatt et al., 2024) and BRO (Nauman et al., 2024) construct larger critic networks and apply a suite of regularization techniques to stabilize training. MEow (Chao et al., 2024) employs energy-based normalizing flows as unified policies to represent both the actor and the critic, simplifying the training process for MaxEnt RL. This paper highlights the importance of policy representation within the MaxEnt RL framework: a more expressive policy representation enhances exploration and facilitates closer convergence to the optimal MaxEnt policy. Diffusion models, which are more expressive than Gaussian distributions and energy-based normalizing flows and easier to train and sample than energy-based models, present an ideal policy representation that effectively balances expressiveness and the complexity of training and inference.

**Diffusion Policies for Offline RL.** Offline RL attempts to learn a well-performing policy from a pre-collected dataset. Collected by multiple policies, the offline datasets may exhibit high skewness and multi-modality. Diffusion Policy (Chi et al., 2023) trains a diffusion model to approximate the multi-modal expert behavior by behavior cloning. To optimize the policy for higher performance, Diffusion-QL (Wang et al., 2023) combines the diffusion loss with Q-value loss evaluated on the generated actions, CEP (Lu et al., 2023) trains a separate guidance network using Q-function to guide the actions to regions with high Q values, and EDA (Chen et al., 2024b) employs direct preference optimization to align the diffusion policy with Q-function. To improve the training and inference speed of diffusion policy, EDP (Kang et al., 2024) adopts action approximation and efficient ODE sampler DPM-solver for action generation, and CPQL (Chen et al., 2024c) utilizes the consistency policy (Song et al., 2023), a one-step diffusion policy. Due to the lack of online samples, the above approaches require staying close to the behavior policy to prevent out-of-distribution actions whose performances are unpredictable. However, in this paper, we focus on online RL, where online interactions are accessible to correct the errors in value evaluation. Therefore, different techniques should be developed to employ diffusion models in online RL.

**Diffusion Policies for Online RL.** In online RL, a key challenge lies in balancing exploration and exploitation. Previous studies (Psenka et al., 2024; Yang et al., 2023; Ding et al., 2024; Wang et al., 2024) apply expressive diffusion models as policy representations to promote the exploration of the state-action space. QSM (Psenka et al., 2024) fits the exponential of the Q-function by training a score network to approximate the action gradient of the Q-function. DIPO (Yang et al., 2023) improves the actions by applying the action gradient of the Q-function and clones the improved actions. QVPO (Ding et al., 2024) weights the diffusion loss with the Q-value, assigning probabilities to actions that

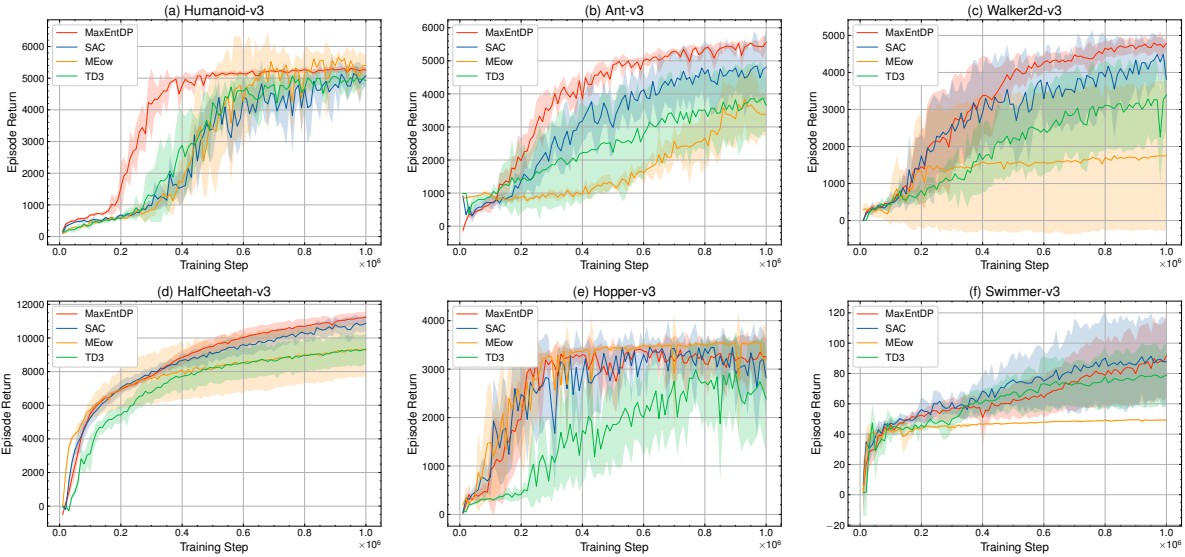

*Figure 1.* Learning curves on Mujoco benchmarks. The solid lines are the means and the shaded regions represent the standard deviations over five runs.

are linearly proportional to the Q-value. DACER (Wang et al., 2024) optimizes the Q-value loss to generate actions with high Q values and adds extra noise to the generated actions to keep a constant policy entropy. Unlike previous approaches, we employ the MaxEnt RL objective to encourage exploration and enhance policy robustness. Similar to QSM, we train the diffusion model to fit the exponential of the Q-function. However, our Q-weighted noise estimation method is more accurate and stable. Furthermore, we include policy entropy when computing the Q-function, which can further promote exploration.

## 5. Experiments

In this section, we conduct experiments to address the following questions: (1) Can MaxEntDP effectively learn a multi-modal policy in a multi-goal task? (2) Does the diffusion policy outperform the Gaussian policy and other generative models within the MaxEnt RL framework? (3) How does performance vary when replacing the Q-weighted Noise Estimation method with competing approaches, such as QSM and iDEM? (4) How does MaxEntDP compare to other diffusion-based online RL algorithms? (5) Does the MaxEnt RL objective benefit policy training?

### 5.1. A Toy Multi-goal Environment

In this subsection, we use a 2-D multi-goal environment (Haarnoja et al., 2017) to demonstrate the effectiveness of MaxEntDP. In this environment, the agent is a 2-D point mass trying to reach one of four symmetrically placed goals. The state and action are position and velocity, respectively. And the reward is a penalty for the velocity and distance

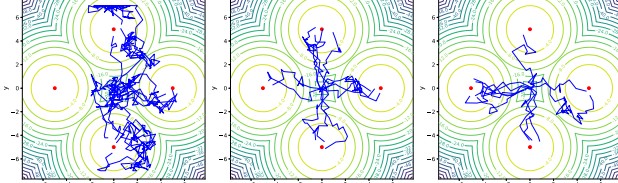

*Figure 2.* The generated trajectories during the training process. From left to right are trajectories generated after 2k, 4k, and 6k training steps. The goals are denoted by the red points.

from the closest goal. Under the MaxEnt RL objective, the optimal policy is to choose one goal randomly and then move toward it. Figure 2 shows the trajectories generated by the diffusion policy during the training process. We can see that MaxEntDP can effectively explore the state-action space and learn a multi-modal policy that approaches the optimal MaxEnt policy.

### 5.2. Comparative Evaluation

**Policy Representations.** To reveal the superiority of applying diffusion models as policy representations to achieve the MaxEnt RL objective, we compare the performance of MaxEntDP on Mujoco benchmarks (Todorov et al., 2012) with other algorithms. Our chosen baseline algorithms include SAC (Haarnoja et al., 2018), MEow (Chao et al., 2024), and TD3 (Fujimoto et al., 2018). SAC and MEow are two methods to pursue the same MaxRnt RL objective using Gaussian policy and energy-based normalizing flow policy, and TD3 provides a contrast to the deterministic policy. Figure 1 shows that MaxEntDP surpasses (a-d) or matches (e-f) baseline algorithms on all tasks, and its evaluation variance

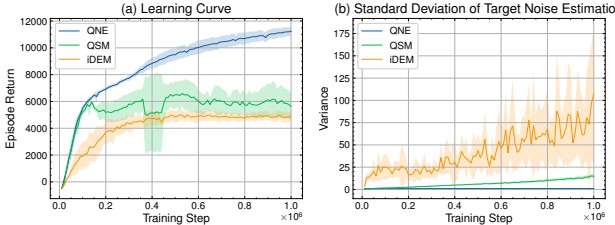

*Figure 3.* Comparison between QNE and two competing methods, QSM and iDEM on HalfCheetah-v3 benchmark. (a) Learning curves. (b) Standard deviations of target noise (a.k.a. the scaled score function) estimation computed on a batch of noisy actions.

is much smaller than other algorithms. This result indicates that the combination of MaxEntRL and diffusion policy effectively balances exploration and exploitation, enabling rapid convergence to a robust and high-performing policy.

**Diffusion Models Training Methods.** In this subsection, we demonstrate the advantages of the proposed Q-weighted Noise Estimation method (QNE) on training diffusion models, compared to two competing methods, QSM and iDEM. We replace the QNE module with QSM and iDEM to observe performance differences. As shown in Figure 3(a), the performance of QSM and iDEM improves initially but then fluctuates after reaching a high level. This may be due to both methods relying on the gradient computation of the Q-function to estimate the score function. When the Q-value is large, its gradient typically varies much across different actions, leading to a high variance in score function estimation for QSM and iDEM, as illustrated in Figure 3(b). This increased variance causes instability in the training of the noise prediction network. In contrast, QNE exhibits significantly lower variance, and its performance improves steadily throughout the training process.

**Diffusion-based Online RL Algorithms.** We also compare MaxEntDP with state-of-the-art diffusion-based online RL algorithms: QSM, DIPO, QVPO, and DACER. These algorithms adopt different techniques to seek a balance between exploration and exploitation. Since the performances of different exploration strategies depend quite a lot on the characteristics of the RL tasks, none of the competing methods performs consistently well on all tasks, as shown by Figure 4. In contrast, our MaxEntDP outperforms or performs comparably to the top method on each task, showing consistent sample efficiency and stability.

### 5.3. Ablation Analysis

In addition, we analyze the function of the MaxEnt RL objective by removing the probability approximation module in MaxEntDP. After doing this, we compute the original Q-function rather than the soft Q-function in the policy evaluation step. As shown in Figure 5, the performance decreases and exhibits greater variance after excluding policy

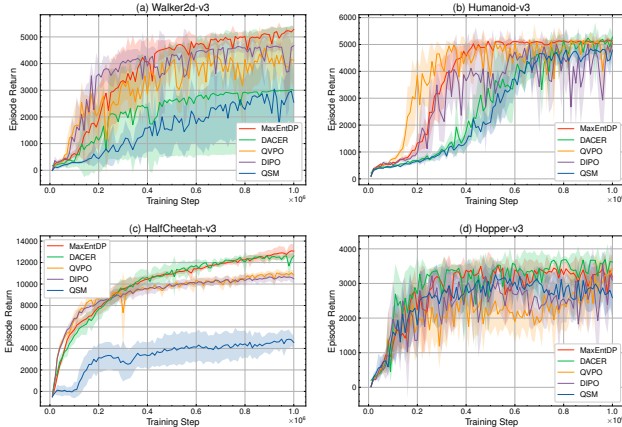

*Figure 4.* Learning curves of diffusion-based online RL algorithms.

entropy in the Q-function. This implies that the MaxEnt RL objective can benefit policy learning: it not only encourages the action distribution at the current step to be more stochastic (by fitting the exponential of Q-function), but also encourages transferring to the states with higher entropy (by computing the soft Q-function). Therefore, the MaxEnt RL objective shows a stronger exploration ability of the whole state-action space, leading to an efficient and stable training process.

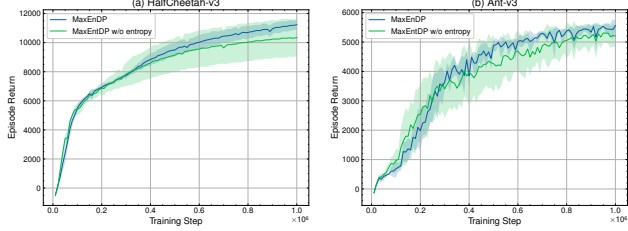

*Figure 5.* The learning curve of MaxEntDP with and without entropy in Q-function computation.

### 6. Conclusion

This paper proposes MaxEntDP, a method to achieve the MaxEnt RL objective with diffusion policies. Compared to the Gaussian policy, the diffusion policy shows stronger exploration ability and expressiveness to approach the optimal MaxEnt policy. To address challenges in applying diffusion policies, we propose Q-weighted noise estimation to train the diffusion model and introduce the numerical integration technique to approximate the probability of diffusion policy. Experiments on Mujoco benchmarks demonstrate that MaxEntDP outperforms Gaussian policy and other generative models within the MaxEnt RL framework, and performs comparably to other diffusion-based online RL algorithms.

**Limitations and Future Work.** Since different RL tasks require varying levels of exploration, we adjust the temperature coefficient for each task and keep it fixed during

training. Future work will explore how to automatically adapt this parameter, making MaxEntDP easier to apply in real-world applications.

## Acknowledgements

We extend our gratitude to the ICML reviewers who evaluated MaxEntDP for their valuable insights and feedback. This work was supported by the National Key Research and Development Program of China (No. 2021ZD0201504).

## Impact Statement

This paper focuses on achieving the MaxEnt RL objective, which is particularly effective for reinforcement learning tasks that require extensive exploration or policy robustness. Beyond advancing RL, our proposed Q-weighted noise estimation and numerical integration techniques address two fundamental issues in diffusion models: fitting the exponential of a given energy function and computing exact probabilities. These two modules can be seamlessly integrated into diffusion-based studies that involve these issues.

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

# A. Theoretic Proofs.

## A.1. Proofs for Soft Actor-Critic Algorithm

Our proof is based on the tabular setting, i.e., $|\mathcal{S}| < \infty$, $|\mathcal{A}| < \infty$ and the replay buffer $\mathcal{D}$ covers all $(\boldsymbol{s}, \boldsymbol{a}) \in |\mathcal{S}| \times |\mathcal{A}|$.

The soft Q-function of policy $\pi$ is defined as:

$$Q^{\pi}(\boldsymbol{s}_t, \boldsymbol{a}_t) = r(\boldsymbol{s}_t, \boldsymbol{a}_t) + \mathbb{E}_{\rho_{\pi}}\left[\sum_{l=1}^{\infty}\gamma^l(r(\boldsymbol{s}_{t+l}, \boldsymbol{a}_{t+l}) - \beta \log \pi(\boldsymbol{a}_{t+l}|\boldsymbol{s}_{t+l}))\right], \tag{22}$$

which satisfies:

$$Q^{\pi}(\boldsymbol{s}_t, \boldsymbol{a}_t) = r(\boldsymbol{s}_t, \boldsymbol{a}_t) + \mathbb{E}_{\rho_{\pi}}\left[\sum_{l=1}^{\infty}\gamma^l(r(\boldsymbol{s}_{t+l}, \boldsymbol{a}_{t+l}) - \beta \log \pi(\boldsymbol{a}_{t+l}|\boldsymbol{s}_{t+l}))\right]$$

$$= r(\boldsymbol{s}_t, \boldsymbol{a}_t) + \mathbb{E}_{\rho_{\pi}}\left[-\gamma\beta \log \pi(\boldsymbol{a}_{t+1}|\boldsymbol{s}_{t+1}) + \gamma r(\boldsymbol{s}_{t+1}, \boldsymbol{a}_{t+1}) + \sum_{l=2}^{\infty}\gamma^l(r(\boldsymbol{s}_{t+l}, \boldsymbol{a}_{t+l}) - \beta \log \pi(\boldsymbol{a}_{t+l}|\boldsymbol{s}_{t+l}))\right]$$
$$\tag{23}$$

$$= r(\boldsymbol{s}_t, \boldsymbol{a}_t) + \gamma\mathbb{E}_{\rho_{\pi}}\left[-\beta \log \pi(\boldsymbol{a}_{t+1}|\boldsymbol{s}_{t+1}) + r(\boldsymbol{s}_{t+1}, \boldsymbol{a}_{t+1}) + \sum_{l=1}^{\infty}\gamma^l(r(\boldsymbol{s}_{t+1+l}, \boldsymbol{a}_{t+1+l}) - \beta \log \pi(\boldsymbol{a}_{t+1+l}|\boldsymbol{s}_{t+1+l}))\right]$$
$$\tag{24}$$

$$= r(\boldsymbol{s}_t, \boldsymbol{a}_t) + \gamma\mathbb{E}_{\boldsymbol{s}_{t+1}\sim p, \boldsymbol{a}_{t+1}\sim\pi}\left[-\beta \log \pi(\boldsymbol{a}_{t+1}|\boldsymbol{s}_{t+1}) + Q^{\pi}(\boldsymbol{s}_{t+1}, \boldsymbol{a}_{t+1})\right]. \tag{25}$$

Equation 25 is called the soft Bellman equation.

**Lemma A.1.** *(Soft Policy Evaluation)* $Q_{\theta}$ *converges to the soft Q-function of* $\pi_{\phi}$ *as* $L(\theta) \to 0$.

*Proof.* Define the soft Bellman operator $\mathcal{T}^{\pi}$ as:

$$\mathcal{T}^{\pi}Q(s, a) = r(\boldsymbol{s}, \boldsymbol{a}) + \gamma\mathbb{E}_{\boldsymbol{s}'\sim p, \boldsymbol{a}'\sim\pi}\left[Q(\boldsymbol{s}', \boldsymbol{a}') - \beta \log \pi(\boldsymbol{a}'|\boldsymbol{s}')\right]. \tag{26}$$

For two Q-function $Q$ and $Q'$, we have

$$|\mathcal{T}^{\pi}Q(\boldsymbol{s}, \boldsymbol{a}) - \mathcal{T}^{\pi}Q'(\boldsymbol{s}, \boldsymbol{a})| = |\gamma\mathbb{E}_{\boldsymbol{s}'\sim p, \boldsymbol{a}'\sim\pi}\left[Q(\boldsymbol{s}', \boldsymbol{a}') - Q'(\boldsymbol{s}', \boldsymbol{a}')\right]| \tag{27}$$

$$\leq \gamma\mathbb{E}_{\boldsymbol{s}'\sim p, \boldsymbol{a}'\sim\pi}\left[|Q(\boldsymbol{s}', \boldsymbol{a}') - Q'(\boldsymbol{s}', \boldsymbol{a}')|\right] \tag{28}$$

$$\leq \gamma\max_{(\boldsymbol{s}', \boldsymbol{a}')}|Q(\boldsymbol{s}', \boldsymbol{a}') - Q'(\boldsymbol{s}', \boldsymbol{a}')| \tag{29}$$

$$= \gamma \parallel Q - Q' \parallel_{\infty} \tag{30}$$

Then

$$\parallel \mathcal{T}^{\pi}Q - \mathcal{T}^{\pi}Q' \parallel_{\infty} \leq \gamma \parallel Q - Q' \parallel_{\infty}, \tag{31}$$

which proves that the soft Bellman operator $\mathcal{T}^{\pi}$ is a contraction. It has a unique fixed point $Q^{\pi}$. When Q-function loss $L(\theta) = 0$, the Q-function $Q_{\theta}$ satisfies the soft Bellman equation $Q_{\theta}(\boldsymbol{s}, \boldsymbol{a}) = r(\boldsymbol{s}, \boldsymbol{a}) + \gamma\mathbb{E}_{\boldsymbol{s}'\sim p, \boldsymbol{a}'\sim\pi_{\phi}}\left[Q_{\theta}(\boldsymbol{s}', \boldsymbol{a}') - \beta \log \pi_{\phi}(\boldsymbol{a}'|\boldsymbol{s}')\right]$ for all $(\boldsymbol{s}, \boldsymbol{a}) \in |\mathcal{S}| \times |\mathcal{A}|$, indicating that $Q_{\theta}$ converges to the true soft function of $\pi_{\phi}$. □

**Lemma A.2.** *(Soft Policy Improvement)* Let $\pi_{\phi_k} \in \Pi$ and assume $Q_{\theta} = Q^{\pi_{\phi_k}}$ after soft policy evaluation. If $\pi_{\phi_{k+1}}$ is the minimizer of the loss defined in Equation 3, then $Q^{\pi_{\phi_{k+1}}}(\boldsymbol{s}, \boldsymbol{a}) \geq Q^{\pi_{\phi_k}}(\boldsymbol{s}, \boldsymbol{a})$ for all $(\boldsymbol{s}, \boldsymbol{a}) \in \mathcal{S} \times \mathcal{A}$ with $|\mathcal{A}| < \infty$.

*Proof.* Since the new policy $\pi_{\pi_{k+1}}$ is the minimizer of the loss defined in Equation 3, it holds that

$$\pi_{\phi_{k+1}}(\cdot|\boldsymbol{s}) = \arg\min_{\pi\in\Pi} D_{KL}\left(\pi(\cdot|\boldsymbol{s})\;\middle\|\;\frac{\exp(\frac{1}{\beta}Q_\theta(\boldsymbol{s},\cdot))}{Z_\theta(\boldsymbol{s})}\right) \tag{32}$$

$$= \arg\min_{\pi\in\Pi} D_{KL}\left(\pi(\cdot|\boldsymbol{s})\;\middle\|\;\frac{\exp(\frac{1}{\beta}Q^{\pi_{\phi_k}}(\boldsymbol{s},\cdot))}{Z^{\pi_{\phi_k}}(\boldsymbol{s})}\right) \tag{33}$$

$$= \arg\min_{\pi\in\Pi}\left\{\mathbb{E}_{\boldsymbol{a}\sim\pi(\cdot|\boldsymbol{s})}\left[\log\pi(\boldsymbol{a}|\boldsymbol{s}) - \frac{1}{\beta}Q^{\pi_{\phi_k}}(\boldsymbol{s},\boldsymbol{a}) + \log Z^{\pi_{\phi_k}}(\boldsymbol{s})\right]\right\} \tag{34}$$

$$= \arg\min_{\pi\in\Pi}\left\{\mathbb{E}_{\boldsymbol{a}\sim\pi(\cdot|\boldsymbol{s})}\left[\log\pi(\boldsymbol{a}|\boldsymbol{s}) - \frac{1}{\beta}Q^{\pi_{\phi_k}}(\boldsymbol{s},\boldsymbol{a})\right] + \log Z^{\pi_{\phi_k}}(\boldsymbol{s})\right\} \tag{35}$$

$$= \arg\min_{\pi\in\Pi}\left\{\mathbb{E}_{\boldsymbol{a}\sim\pi(\cdot|\boldsymbol{s})}\left[\log\pi(\boldsymbol{a}|\boldsymbol{s}) - \frac{1}{\beta}Q^{\pi_{\phi_k}}(\boldsymbol{s},\boldsymbol{a})\right]\right\}. \tag{36}$$

Since $\pi_{\phi_k}\in\Pi$, we have

$$\mathbb{E}_{\boldsymbol{a}\sim\pi_{\phi_{k+1}}(\cdot|\boldsymbol{s})}\left[\log\pi_{\phi_{k+1}}(\boldsymbol{a}|\boldsymbol{s}) - \frac{1}{\beta}Q^{\pi_{\phi_k}}(\boldsymbol{s},\boldsymbol{a})\right] \le \mathbb{E}_{\boldsymbol{a}\sim\pi_{\phi_k}(\cdot|\boldsymbol{s})}\left[\log\pi_{\phi_k}(\boldsymbol{a}|\boldsymbol{s}) - \frac{1}{\beta}Q^{\pi_{\phi_k}}(\boldsymbol{s},\boldsymbol{a})\right]. \tag{37}$$

According the soft Bellman equation, the Q-function of $\pi_{\phi_k}$ satisfies

$$Q^{\pi_{\phi_k}}(\boldsymbol{s}_t,\boldsymbol{a}_t) = r(\boldsymbol{s}_t,\boldsymbol{a}_t) + \gamma\mathbb{E}_{\boldsymbol{s}_{t+1}\sim p,\boldsymbol{a}_{t+1}\sim\pi_{\phi_k}}[Q^{\pi_{\phi_k}}(\boldsymbol{s}_{t+1},\boldsymbol{a}_{t+1}) - \beta\log\pi_{\phi_k}(\boldsymbol{a}_{t+1}|\boldsymbol{s}_{t+1})] \tag{38}$$

$$\le r(\boldsymbol{s}_t,\boldsymbol{a}_t) + \gamma\mathbb{E}_{\boldsymbol{s}_{t+1}\sim p,\boldsymbol{a}_{t+1}\sim\pi_{\phi_{k+1}}}\left[Q^{\pi_{\phi_k}}(\boldsymbol{s}_{t+1},\boldsymbol{a}_{t+1}) - \beta\log\pi_{\phi_{k+1}}(\boldsymbol{a}_{t+1}|\boldsymbol{s}_{t+1})\right] \tag{39}$$

$$= r(\boldsymbol{s}_t,\boldsymbol{a}_t) + \gamma\mathbb{E}_{\boldsymbol{s}_{t+1}\sim p,\boldsymbol{a}_{t+1}\sim\pi_{\phi_{k+1}}}\left[r(\boldsymbol{s}_{t+1},\boldsymbol{a}_{t+1}) - \beta\log\pi_{\phi_{k+1}}(\boldsymbol{a}_{t+1}|\boldsymbol{s}_{t+1})\right] \tag{40}$$

$$+ \gamma^2\mathbb{E}_{\boldsymbol{s}_{t+1}\sim p,\boldsymbol{a}_{t+1}\sim\pi_{\phi_{k+1}},\boldsymbol{s}_{t+2}\sim p,\boldsymbol{a}_{t+2}\sim\pi_{\phi_k}}[Q^{\pi_{\phi_k}}(\boldsymbol{s}_{t+2},\boldsymbol{a}_{t+2}) - \beta\log\pi_{\phi_k}(\boldsymbol{a}_{t+2}|\boldsymbol{s}_{t+2})] \tag{41}$$

$$\vdots \tag{42}$$

$$\le Q^{\pi_{\phi_{k+1}}}(\boldsymbol{s}_t,\boldsymbol{a}_t), \tag{43}$$

which is proved by repeatedly expanding $Q^{\pi_{\phi_k}}$ using the soft Bellman equation and applying Equation 37. Then the proof for Lemma A.2 is completed. □

**Theorem A.3.** *(Soft Policy Iteration) In the tabular setting, let $L(\theta_k) = 0$ and $L(\phi_k)$ be minimized for each k. Repeated application of policy evaluation and policy improvement, i.e., $k \to \infty$, $\pi_{\phi_k}$ will converge to a policy $\pi^*$ such that $Q^{\pi^*}(\boldsymbol{s},\boldsymbol{a}) \ge Q^\pi(\boldsymbol{s},\boldsymbol{a})$ for all $\pi \in \Pi$ and $(\boldsymbol{s},\boldsymbol{a}) \in \mathcal{S}\times\mathcal{A}$ with $|\mathcal{A}| < \infty$.*

*Proof.* According to Lemma A.1 and A.2, when $L(\theta_k) = 0$ and $L(\phi_k)$ be minimized for each $k$, we have $\forall k, Q^{\pi_{\phi_{k+1}}} \ge Q^{\pi_{\phi_k}}$. This indicates that the sequence $Q^{\pi_{\phi_k}}$ is monotonically increasing. Furthermore, the Q-function is bounded since both the reward and entropy are bound. Therefore, when $k \to \infty$, the policy sequence converges to some $\pi^*$. Below we will prove that $\pi^*$ is the optimal MaxEnt policy within $\pi$.

Since $\pi^*$ has already converged, it satisfies

$$\pi^*(\cdot|\boldsymbol{s}) = \arg\min_{\pi\in\Pi}\left\{\mathbb{E}_{\boldsymbol{a}\sim\pi(\cdot|\boldsymbol{s})}\left[\log\pi(\boldsymbol{a}|\boldsymbol{s}) - \frac{1}{\beta}Q^{\pi^*}(\boldsymbol{s},\boldsymbol{a})\right]\right\} \tag{44}$$

following Equation 36. Then for all $\pi \in \Pi$, it holds that

$$\mathbb{E}_{\boldsymbol{a}\sim\pi^*(\cdot|\boldsymbol{s})}\left[\log\pi^*(\boldsymbol{a}|\boldsymbol{s}) - \frac{1}{\beta}Q^{\pi^*}(\boldsymbol{s},\boldsymbol{a})\right] \le \mathbb{E}_{\boldsymbol{a}\sim\pi(\cdot|\boldsymbol{s})}\left[\log\pi(\boldsymbol{a}|\boldsymbol{s}) - \frac{1}{\beta}Q^{\pi^*}(\boldsymbol{s},\boldsymbol{a})\right]. \tag{45}$$

Using the same iterative argument as in the proof of Equation 43, we can derive $Q^{\pi^*}(\boldsymbol{s},\boldsymbol{a}) \ge Q^\pi(\boldsymbol{s},\boldsymbol{a})$ for all $(\boldsymbol{s},\boldsymbol{a}) \in \mathcal{S}\times\mathcal{A}$. Consequently, $\pi^*$ is the optimal MaxEnt policy within $\Pi$. The proof is completed. □

### A.2. The Decomposition of the Condition Distribution $p(\boldsymbol{a}_0|\boldsymbol{a}_t)$

According to the Bayesian rule, the conditional distribution $p(\boldsymbol{a}_0|\boldsymbol{a}_t)$ satisfies:

$$p(\boldsymbol{a}_0|\boldsymbol{a}_t) = \frac{p(\boldsymbol{a}_0)p(\boldsymbol{a}_t|\boldsymbol{a_0})}{p(\boldsymbol{x}_t)} \tag{46}$$

$$\propto p(\boldsymbol{a}_0)p(\boldsymbol{a}_t|\boldsymbol{a_0}) \tag{47}$$

$$\propto \exp(\frac{1}{\beta}Q(\boldsymbol{a_0}))\mathcal{N}(\boldsymbol{a}_t|\sqrt{\sigma(\alpha_t)}\boldsymbol{a}_0, \sigma(-\alpha_t)\boldsymbol{I}), \tag{48}$$

where Equation 48 is derived by substituting Equation 9 and 10. For the same $\boldsymbol{a}_0$ and $\boldsymbol{a}_t$, the probability density of $\boldsymbol{a}_t$ following the Gaussian distribution $\mathcal{N}(\boldsymbol{a}_t|\sqrt{\sigma(\alpha_t)}\boldsymbol{a}_0, \sigma(-\alpha_t)\boldsymbol{I})$ is equal to the probability density of $\boldsymbol{a}_0$ following the Gaussian distribution $\mathcal{N}(\boldsymbol{a}_0|\frac{1}{\sqrt{\sigma(\alpha_t)}}\boldsymbol{a}_t, \frac{\sigma(-\alpha_t)}{\sigma(\alpha_t)}\boldsymbol{I})$ up to a constant to compensate for the scale difference between the two random variables. Then we have

$$p(\boldsymbol{a}_0|\boldsymbol{a}_t) \propto \exp(\frac{1}{\beta}Q(\boldsymbol{a_0}))\mathcal{N}(\boldsymbol{a}_0|\frac{1}{\sqrt{\sigma(\alpha_t)}}\boldsymbol{a}_t, \frac{\sigma(-\alpha_t)}{\sigma(\alpha_t)}\boldsymbol{I}). \tag{49}$$

### A.3. Estimating the Score Function with Importance Sampling

The score function satisfies

$$\nabla_{\boldsymbol{a}_t} \log p(\boldsymbol{a}_t) = \mathbb{E}_{p(\boldsymbol{a}_0|\boldsymbol{a}_t)} \left[ \nabla_{\boldsymbol{a}_t} \log p(\boldsymbol{a}_t|\boldsymbol{a}_0) \right] \tag{50}$$

$$= \mathbb{E}_{\boldsymbol{a}_0 \sim \mathcal{N}(\boldsymbol{a}_0|\frac{1}{\sqrt{\sigma(\alpha_t)}}\boldsymbol{a}_t, \frac{\sigma(-\alpha_t)}{\sigma(\alpha_t)}\boldsymbol{I})} \left[ \frac{p(\boldsymbol{a}_0|\boldsymbol{a}_t)}{\mathcal{N}(\boldsymbol{a}_0|\frac{1}{\sqrt{\sigma(\alpha_t)}}\boldsymbol{a}_t, \frac{\sigma(-\alpha_t)}{\sigma(\alpha_t)}\boldsymbol{I})} \nabla_{\boldsymbol{a}_t} \log p(\boldsymbol{a}_t|\boldsymbol{a}_0) \right] \tag{51}$$

$$= \mathbb{E}_{\boldsymbol{a}_0 \sim \mathcal{N}(\boldsymbol{a}_0|\frac{1}{\sqrt{\sigma(\alpha_t)}}\boldsymbol{a}_t, \frac{\sigma(-\alpha_t)}{\sigma(\alpha_t)}\boldsymbol{I})} \left[ w(\boldsymbol{a}_0)\nabla_{\boldsymbol{a}_t} \log p(\boldsymbol{a}_t|\boldsymbol{a}_0) \right] \tag{52}$$

$$= \mathbb{E}_{\boldsymbol{a}_0 \sim \mathcal{N}(\boldsymbol{a}_0|\frac{1}{\sqrt{\sigma(\alpha_t)}}\boldsymbol{a}_t, \frac{\sigma(-\alpha_t)}{\sigma(\alpha_t)}\boldsymbol{I})} \left[ -w(\boldsymbol{a}_0)\frac{\boldsymbol{a}_t - \sqrt{\sigma(\alpha_t)}\boldsymbol{a}_0}{\sigma(-\alpha_t)} \right], \tag{53}$$

where the importance ratio $w(\boldsymbol{a}_0) = \frac{\exp(\frac{1}{\beta}Q(\boldsymbol{a_0}))}{Z(\boldsymbol{a}_t)}$ with $Z(\boldsymbol{a}_t) = \int \exp(\frac{1}{\beta}Q(\boldsymbol{a_0}))\mathcal{N}(\boldsymbol{a}_0|\frac{1}{\sqrt{\sigma(\alpha_t)}}\boldsymbol{a}_t, \frac{\sigma(-\alpha_t)}{\sigma(\alpha_t)}\boldsymbol{I})\mathrm{d}\boldsymbol{a}_0$ being the normalizing constant of $p(\boldsymbol{a}_0|\boldsymbol{a}_t)$. Let $\boldsymbol{a}_0 = \frac{1}{\sqrt{\sigma(\alpha_t)}}\boldsymbol{a}_t + \frac{\sqrt{\sigma(-\alpha_t)}}{\sqrt{\sigma(\alpha_t)}}\boldsymbol{\epsilon}$, then Equation 53 can be rewritten as

$$\nabla_{\boldsymbol{a}_t} \log p(\boldsymbol{a}_t) = \frac{1}{\sqrt{\sigma(-\alpha_t)}} \cdot \mathbb{E}_{\boldsymbol{\epsilon} \sim \mathcal{N}(\boldsymbol{0}, \boldsymbol{I})} \left[ w(\boldsymbol{a}_0)\boldsymbol{\epsilon} \right] \tag{54}$$

$$\approx \frac{1}{\sqrt{\sigma(-\alpha_t)}} \cdot \frac{1}{K}\sum_{i=1}^{K} w(\boldsymbol{a}_0^i)\boldsymbol{\epsilon}^i, \tag{55}$$

where $\boldsymbol{\epsilon}^1, \ldots, \boldsymbol{\epsilon}^K \sim \mathcal{N}(\boldsymbol{0}, \boldsymbol{I})$, $\boldsymbol{a}_0^i = \frac{1}{\sqrt{\sigma(\alpha_t)}}\boldsymbol{a}_t + \frac{\sqrt{\sigma(-\alpha_t)}}{\sqrt{\sigma(\alpha_t)}}\boldsymbol{\epsilon}^i$.

### A.4. The Derivation of the iDEM Method

We provide the derivation of the iDEM method to demonstrate the difference and relationship between Q-weighted noise estimation and iDEM. Our derivation is equivalent to the official proof of iDEM, although in a different way.

Since $\nabla_{\boldsymbol{a}_0} \log \mathcal{N}(\boldsymbol{a}_0|\frac{1}{\sqrt{\sigma(\alpha_t)}}\boldsymbol{a}_t, \frac{\sigma(-\alpha_t)}{\sigma(\alpha_t)}\boldsymbol{I}) = \sqrt{\sigma(\alpha_t)} \cdot \frac{\boldsymbol{a}_t - \sqrt{\sigma(\alpha_t)}\boldsymbol{a}_0}{\sigma(-\alpha_t)}$ and $\nabla_{\boldsymbol{a}_t} \log p(\boldsymbol{a}_t|\boldsymbol{a}_0) = -\frac{\boldsymbol{a}_t - \sqrt{\sigma(\alpha_t)}\boldsymbol{a}_0}{\sigma(-\alpha_t)}$, it holds that

$$\nabla_{\boldsymbol{a}_t} \log p(\boldsymbol{a}_t|\boldsymbol{a}_0) = -\frac{1}{\sqrt{\sigma(\alpha_t)}}\nabla_{\boldsymbol{a}_0} \log \mathcal{N}(\boldsymbol{a}_0|\frac{1}{\sqrt{\sigma(\alpha_t)}}\boldsymbol{a}_t, \frac{\sigma(-\alpha_t)}{\sigma(\alpha_t)}\boldsymbol{I}). \tag{56}$$

Substitute Equation 56 into Equation 52, we have

$$\nabla_{\boldsymbol{a}_t} \log p(\boldsymbol{a}_t) = -\frac{1}{\sqrt{\sigma(\alpha_t)}} \mathbb{E}_{\boldsymbol{a}_0 \sim \mathcal{N}(\boldsymbol{a}_0 | \frac{1}{\sqrt{\sigma(\alpha_t)}} \boldsymbol{a}_t, \frac{\sigma(-\alpha_t)}{\sigma(\alpha_t)} \boldsymbol{I})} \left[ w(\boldsymbol{a}_0) \nabla_{\boldsymbol{a}_0} \log \mathcal{N}(\boldsymbol{a}_0 | \frac{1}{\sqrt{\sigma(\alpha_t)}} \boldsymbol{a}_t, \frac{\sigma(-\alpha_t)}{\sigma(\alpha_t)} \boldsymbol{I}) \right] \quad (57)$$

$$= -\frac{1}{\sqrt{\sigma(\alpha_t)}} \int w(\boldsymbol{a}_0) \nabla_{\boldsymbol{a}_0} \mathcal{N}(\boldsymbol{a}_0 | \frac{1}{\sqrt{\sigma(\alpha_t)}} \boldsymbol{a}_t, \frac{\sigma(-\alpha_t)}{\sigma(\alpha_t)} \boldsymbol{I}) \mathrm{d}\boldsymbol{a}_0. \quad (58)$$

After applying the integration by parts formula, Equation 58 can be expanded to

$$\nabla_{\boldsymbol{a}_t} \log p(\boldsymbol{a}_t) = \frac{1}{\sqrt{\sigma(\alpha_t)}} \int (\nabla_{\boldsymbol{a}_0} w(\boldsymbol{a}_0)) \cdot \mathcal{N}(\boldsymbol{a}_0 | \frac{1}{\sqrt{\sigma(\alpha_t)}} \boldsymbol{a}_t, \frac{\sigma(-\alpha_t)}{\sigma(\alpha_t)} \boldsymbol{I}) \mathrm{d}\boldsymbol{a}_0. \quad (59)$$

Since $w(\boldsymbol{a}_0) = \frac{\exp(\frac{1}{\beta} Q(\boldsymbol{a}_0))}{Z(\boldsymbol{a}_t)}$, it satisfies that $\nabla_{\boldsymbol{a}_0} w(\boldsymbol{a}_0) = w(\boldsymbol{a}_0) \nabla_{\boldsymbol{a}_0} \frac{1}{\beta} Q(\boldsymbol{a}_0)$. Then we have

$$\nabla_{\boldsymbol{a}_t} \log p(\boldsymbol{a}_t) = \frac{1}{\sqrt{\sigma(\alpha_t)}} \int w(\boldsymbol{a}_0) \nabla_{\boldsymbol{a}_0} \frac{1}{\beta} Q(\boldsymbol{a}_0) \cdot \mathcal{N}(\boldsymbol{a}_0 | \frac{1}{\sqrt{\sigma(\alpha_t)}} \boldsymbol{a}_t, \frac{\sigma(-\alpha_t)}{\sigma(\alpha_t)} \boldsymbol{I}) \mathrm{d}\boldsymbol{a}_0 \quad (60)$$

$$= \frac{1}{\sqrt{\sigma(\alpha_t)}} \mathbb{E}_{\boldsymbol{a}_0 \sim \mathcal{N}(\boldsymbol{a}_0 | \frac{1}{\sqrt{\sigma(\alpha_t)}} \boldsymbol{a}_t, \frac{\sigma(-\alpha_t)}{\sigma(\alpha_t)} \boldsymbol{I})} \left[ w(\boldsymbol{a}_0) \nabla_{\boldsymbol{a}_0} \frac{1}{\beta} Q(\boldsymbol{a}_0) \right] \quad (61)$$

Equation 61 appears similar to Equation 53, except that the random variable in the expectation transfers from Q-weighted noise to Q-weighted gradient. Utilizing the same weighted importance sampling method as Q-weighted noise, the score function can be estimated by

$$\nabla_{\boldsymbol{a}_t} \log p(\boldsymbol{a}_t) \approx \frac{1}{\sqrt{\sigma(\alpha_t)}} \cdot \sum_{i=1}^{K} \frac{w(\boldsymbol{a}_0^i)}{\sum_{j=1}^{K} w(\boldsymbol{a}_0^j)} \nabla_{\boldsymbol{a}_0^i} \frac{1}{\beta} Q(\boldsymbol{a}_0^i) \quad (62)$$

$$= \frac{1}{\sqrt{\sigma(\alpha_t)}} \sum_{i=1}^{K} \mathrm{softmax}(\frac{1}{\beta} Q(\boldsymbol{a}_0^{1:K}))_i \nabla_{\boldsymbol{a}_0^i} \frac{1}{\beta} Q(\boldsymbol{a}_0^i). \quad (63)$$

## A.5. Probabity Approximation of Diffusion Policy Using Numerical Integration Techniques

We use numerical integration techniques to estimate the following integral:

$$\log p_\phi(\boldsymbol{a}_0) = c - \frac{1}{2} \int_{-\infty}^{+\infty} \mathbb{E}_{\boldsymbol{\epsilon}} \left[ \| \boldsymbol{\epsilon} - \boldsymbol{\epsilon}_\phi(\boldsymbol{a}_t, \alpha_t) \|_2^2 \right] \mathrm{d}\alpha_t, \quad (64)$$

where $c = -\frac{d}{2} \log(2\pi e) + \frac{d}{2} \int_{-\infty}^{+\infty} \sigma(\alpha_t) \mathrm{d}\alpha_t$ with $d$ being the dimension of $\boldsymbol{a}_0$, $\boldsymbol{\epsilon} \sim \mathcal{N}(\boldsymbol{0}, \boldsymbol{I})$, $\boldsymbol{a}_t = \sqrt{\sigma(\alpha_t)} \boldsymbol{a}_0 + \sqrt{\sigma(-\alpha_t)} \boldsymbol{\epsilon}$. First, using the equation $\alpha_t = \log \frac{\sigma(\alpha_t)}{1 - \sigma(\alpha_t)}$, we change the integral variable from $\alpha_t$ to $\sigma(\alpha_t)$ as $\sigma(\alpha_t)$ has a narrow integration domain of $(0, 1)$:

$$\log p_\phi(\boldsymbol{a}_0) = -\frac{d}{2} \log(2\pi e) + \frac{1}{2} \int_0^1 \left( d \cdot \sigma(\alpha_t) - \mathbb{E}_{\boldsymbol{\epsilon}} \left[ \| \boldsymbol{\epsilon} - \boldsymbol{\epsilon}_\phi(\boldsymbol{a}_t, \alpha_t) \|_2^2 \right] \right) \frac{\mathrm{d}\sigma(\alpha_t)}{\sigma(\alpha_t)\sigma(-\alpha_t)}. \quad (65)$$

In practice, we calculate the integral between $[\sigma(\alpha_{t_{\max}}), \sigma(\alpha_{t_{\min}})]$ for numerical stability, where in our experiments, $t_{\min} = 1e-3$ and $t_{\max} = 0.9946$. Obtain $T+1$ discrete timesteps by setting $t_i = t_{\min} + \frac{i}{T}(t_{\max} - t_{\min})$, $i = 0, 1, \ldots, T$. Then the integration domain of $[\sigma(\alpha_{t_{\max}}), \sigma(\alpha_{t_{\min}})]$ is split into $T$ intervals, where the range of the $i$-th segment is $[\sigma(\alpha_{t_i}), \sigma(\alpha_{t_{i-1}})]$. Using the left-hand endpoints to represent the function value of each interval, the integral can be approximated by

$$\log p_\phi(\boldsymbol{a}_0) \approx -\frac{d}{2} \log(2\pi e) + \frac{1}{2} \sum_{i=1}^{T} \left( d \cdot \sigma(\alpha_{t_i}) - \mathbb{E}_{\boldsymbol{\epsilon}} \left[ \| \boldsymbol{\epsilon} - \boldsymbol{\epsilon}_\phi(\boldsymbol{a}_{t_i}, \alpha_{t_i}) \|_2^2 \right] \right) \frac{\sigma(\alpha_{t_{i-1}}) - \sigma(\alpha_{t_i})}{\sigma(\alpha_{t_i})\sigma(-\alpha_{t_i})}. \quad (66)$$

Estimating the noise predicting error $\mathbb{E}_{\boldsymbol{\epsilon}} \left[ \| \boldsymbol{\epsilon} - \boldsymbol{\epsilon}_\phi(\boldsymbol{a}_{t_i}, \alpha_{t_i}) \|_2^2 \right]$ using Monte Carlo samples, we have

$$\log p_\phi(\boldsymbol{a}_0) \approx -\frac{d}{2} \log(2\pi e) + \frac{1}{2} \sum_{i=1}^{T} \left( d \cdot \sigma(\alpha_{t_i}) - \frac{1}{N} \sum_{j=1}^{N} \| \boldsymbol{\epsilon}^j - \boldsymbol{\epsilon}_\phi(\boldsymbol{a}_{t_i}^j, \alpha_{t_i}) \|_2^2 \right) \frac{\sigma(\alpha_{t_{i-1}}) - \sigma(\alpha_{t_i})}{\sigma(\alpha_{t_i})\sigma(-\alpha_{t_i})}, \quad (67)$$

where $\boldsymbol{\epsilon}^1, \ldots, \boldsymbol{\epsilon}^N \sim \mathcal{N}(\boldsymbol{0}, \boldsymbol{I})$, $\boldsymbol{a}_{t_i}^j = \sqrt{\sigma(\alpha_{t_i})}\boldsymbol{a}_0 + \sqrt{\sigma(-\alpha_{t_i})}\boldsymbol{\epsilon}^j$. The equation 67 can be short for

$$\log p_\phi(\boldsymbol{a}_0) \approx c' + \frac{1}{2}\sum_{i=1}^{T} w_{t_i}\left(d \cdot \sigma(\alpha_{t_i}) - \tilde{\boldsymbol{\epsilon}}_\phi(\boldsymbol{a}_{t_i}, \alpha_{t_i})\right) \tag{68}$$

where $c' = -\frac{d}{2}\log(2\pi e)$, $w_{t_i} = \frac{\sigma(\alpha_{t_{i-1}}) - \sigma(\alpha_{t_i})}{\sigma(\alpha_{t_i})\sigma(-\alpha_{t_i})}$ is the weight at $t_i$, $\tilde{\boldsymbol{\epsilon}}_\phi(\boldsymbol{a}_{t_i}, \alpha_{t_i}) = \frac{1}{N}\sum_{j=1}^{N} \parallel \boldsymbol{\epsilon}^j - \boldsymbol{\epsilon}_\phi(\boldsymbol{a}_{t_i}^j, \alpha_{t_i}) \parallel_2^2$ is the noise prediction error estimation at $t_i$.

## B. Supplementary Related Work on Diffusion-based Energy Models

A line of work focuses on applying the expressive diffusion models to approximate the exponential of a given energy function. QSM (Psenka et al., 2024) trains the score function by aligning it with the gradient of the energy function. iDEM (Akhound-Sadegh et al., 2024) proposes a weighted sum of the gradient of the energy function to estimate the true score function. These two approaches, which are based on gradient computation, suffer from a large estimation variance and demonstrate training instability when used for diffusion policy optimization, as evidenced in Section 5.2. Recently, a work (Sendera et al., 2024) considers the Euler-Maruyama samplers of diffusion models as continuous generative flow networks (GFlowNets), and exploits the trajectory balance objective to train diffusion models. In this method, a replay buffer is used to store the sample generation trajectories of diffusion models, which may cause a heavy memory burden. The model-based diffusion (Pan et al., 2024) proposes the Monte Carlo estimation for computing the score function and uses the Monte Carlo score ascent to generate samples following the Boltzmann distribution of a given function. The model-based diffusion is similar to our QNE method, however, QNE has several properties that matter in RL training:

- We use a parameterized network to approximate the scaled score function, while the model-based diffusion needs to compute the score function using Monte Carlo estimation when generating samples. Therefore, sample generation of model-based diffusion is time-consuming, which will slow the training speed of the RL algorithms.

- We adopt ancestral sampling in DDPM to generate samples, that are more diverse than that of Monte Carlo score ascent used in model-based diffusion.

- We propose to modify the standard Gaussian to the truncated Gaussian in QNE to model the action distribution with a bounded action space. However, model-based diffusion can not address such a bounded distribution.

These properties make QNE well-suited for the optimization of diffusion policy.

## C. Experimental Details

### C.1. Hyperparameters Settings

All experiments in this paper are conducted on a GPU of Nvidia GeForce RTX 3090 and a CPU of AMD EPYC 7742. Our implementation of SAC, MEow, TD3, QSM, DACER, QVPO, and DIPO follows their official codes: https://github.com/toshikwa/soft-actor-critic.pytorch, https://github.com/ChienFeng-hub/meow, https://github.com/sfujim/TD3, https://github.com/Alescontrela/score_matching_rl, https://github.com/happy-yan/DACER-Diffusion-with-Online-RL, https://github.com/wadx2019/qvpo, and https://github.com/BellmanTimeHut/DIPO. The shared hyperparameters of all algorithms are listed in Table 1[2].

### C.2. Training Time

The training time for all algorithms is presented in Table 2. Leveraging the computation efficiency of JAX (Frostig et al., 2018) and the parallel processing capabilities of GPU, MaxEntDP demonstrates high training efficiency compared to competing methods, only behind TD3 and QSM. This highlights its advantage for real-world applications that require high computation efficiency.

---

[2]When comparing with other diffusion-based algorithms, MaxEntDP uses 3-layer MLPs as the actor and critic networks following the default settings of these algorithms. In other experiments, 2-layer MLPs are used as they can already attain good performance.

*Table 1.* The shared hyperparameters of all algorithms.

| Hyperparameter | MaxEntDP | SAC | MEow | TD3 | QSM | DACER | QVPO | DIPO |
|---|---|---|---|---|---|---|---|---|
| Batch size | 256 | 256 | 256 | 256 | 256 | 256 | 256 | 256 |
| Discount factor $\gamma$ | 0.99 | 0.99 | 0.99 | 0.99 | 0.99 | 0.99 | 0.99 | 0.99 |
| Target smoothing coefficient $\tau$ | 0.005 | 0.005 | 0.005 | 0.005 | 0.005 | 0.005 | 0.005 | 0.005 |
| No. of hidden layers | 2 | 2 | 2 | 2 | 3 | 3 | 3 | 3 |
| No. of hidden nodes | 256 | 256 | 256 | 256 | 256 | 256 | 256 | 256 |
| Actor learning rate | 3e-4 | 3e-4 | 3e-4 | 3e-4 | 3e-4 | 3e-4 | 3e-4 | 3e-4 |
| Critic learning rate | 3e-4 | 3e-4 | 3e-4 | 3e-4 | 3e-4 | 3e-4 | 3e-4 | 3e-4 |
| Activation | mish | relu | relu | relu | mish | mish | mish | mish |
| Replay buffer size | 1e6 | 1e6 | 1e6 | 1e6 | 1e6 | 1e6 | 1e6 | 1e6 |
| Diffusion steps | 20 | N/A | N/A | N/A | 20 | 20 | 20 | 20 |
| Action candidate number | 10 | N/A | N/A | N/A | N/A | N/A | 32 | N/A |

*Table 2.* The comparison of training time on HalfCheetah-v3 benchmark.

| Algorithm (2-layer MLP) | MaxEntDP (jax) | SAC | MEow | TD3 |
|---|---|---|---|---|
| Training time (h) | 3.9 | 4.6 | 11 | 1.7 |

| Algorithm (3-layer MLP) | MaxEntDP (jax) | QSM (jax) | DACER (jax) | QVPO | DIPO |
|---|---|---|---|---|---|
| Training time (h) | 5.5 | 1.9 | 5.9 | 22.6 | 55.6 |

## D. Supplementary Experiments

### D.1. Hyperparameter Analysis

In this subsection, we analyze the effect of different hyperparameter settings on the performance:

- **Sample Number for Q-weighted Noise Estimation.** The Q-weighted noise estimation can be seen as a weighted importance sampling method to estimate the training target of the noise prediction network. With more samples, the estimation will be more accurate and less varied, which benefits the training of diffusion policy. This is consistent with the observation in Figure 6(a) that the performance will be better with a larger $K$. We select $K = 500$ since it can obtain good performance and cause relatively small computation costs.

- **Sample Number for Probability Approximation.** For probability approximation of diffusion policy, several Monte Carlo samples are utilized to estimate the noise prediction error at each diffusion timestep. This sample number is also preferred to be large for higher accuracy and less variance. The performance of different sample numbers $N$ is shown in Figure 6(b). We set $N = 50$ after trading off performance and computation efficiency.

- **Diffusion Steps.** Due to the discretization error of ODE solvers, the actual distribution of generated actions may be different from the diffusion policy induced by the noise prediction network. Therefore, when the diffusion steps $T$ is small, the non-negligible discretization error will disrupt the training process and lead to a performance drop. As shown in Figure 6(c), the performance is higher with larger $T$. We choose $T = 20$ as the default setting for a balance between performance and computation efficiency.

- **Candidate Number for Action Selection.** By selecting the action with the highest Q-value among several action candidates, the action selection technique can further improve the performance of the diffusion policy when testing. Figure 6(d) demonstrates that a larger number of action candidates will result in a better performance. Consequently, we set $M = 10$ by default.

- **Temperature Coefficient.** The temperature coefficient $\beta$, which determines the exploration strength, is an important parameter in the MaxEnt RL framework. Since the difficulty and reward scales vary across different tasks, different $\beta$ need to be set for different tasks. We sweep over $[0.01, 0.02, 0.05, 0.1, 0.2]$ to find the optimal setting for each task, displaying the results in Figure 7. The temperature coefficient selected for each task is listed in Table 3.

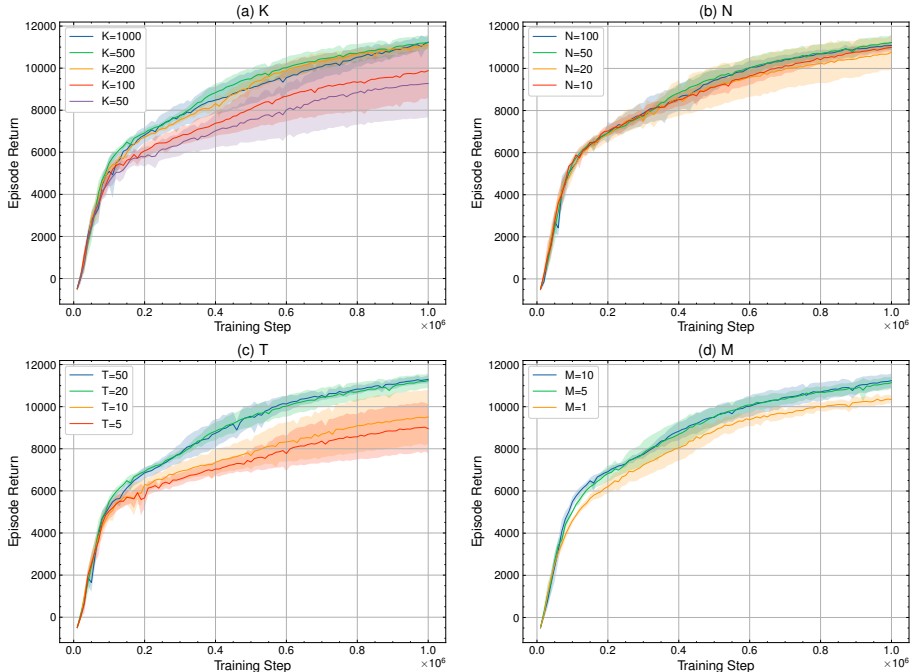

*Figure 6.* Learning curves of different parameter settings on HalfCheetah-v3 benchmark. (a) Testing different numbers of samples $K$ for Q-weighted noise estimation. (b) Testing different numbers of samples $N$ for probability approximation. (c) Testing different diffusion steps $T$. (d) Testing different candidate numbers $M$ for action selection.

*Table 3.* The temperature coefficients adapted for each benchmark.

| Benchmark | Temperature coefficient |
|---|---|
| Ant-v3 | 0.05 |
| HalfCheetah-v3 | 0.2 |
| Hopper-v3 | 0.05 |
| Humanoid-v3 | 0.02 |
| Swimmer-v3 | 0.01 |
| Walker2d-v3 | 0.01 |

### D.2. Multimodal Policy Learning on the Challenging AntMaze Benchmarks

We adopt the AntMaze benchmarks proposed in DDiffPG (Li et al., 2024) to test the multi-modal policy learning ability of MaxEntDP on the challenging high-dimensional RL tasks. In this environment, the agent is a quadruped robot trying to reach the specified goals. Instead of the sparse reward employed in DDiffPG, we use a dense reward, a penalty for the distance from the closest goal, to guide policy learning. We demonstrate the trajectories generated by MaxEntDP and SAC after 1M environment interactions in Figure 8. MaxEntDP can learn diverse behavior modes even in the challenging high-dimensional tasks, while SAC fails to learn different solutions. In addition, we visualize state coverage through the training process for MaxEntDP and SAC, showing the results in Figure 9. We can see that MaxEntDP can explore multiple behavior modes at the same time, while SAC focuses only on a simple mode. This reveals the importance of using the expressive diffusion policy for efficient exploration and multimodal policy learning.

### D.3. Comparative Evaluation on the DeepMind Control Suite

We test MaxEntDP on 3 high-dimensional tasks on the DeepMind Control Suite benchmarks. The performance comparison with SAC, MEow, and TD3 is displayed in Figure 10. MaxEntDP outperforms other baseline algorithms on these challenging high-dimensional RL tasks.

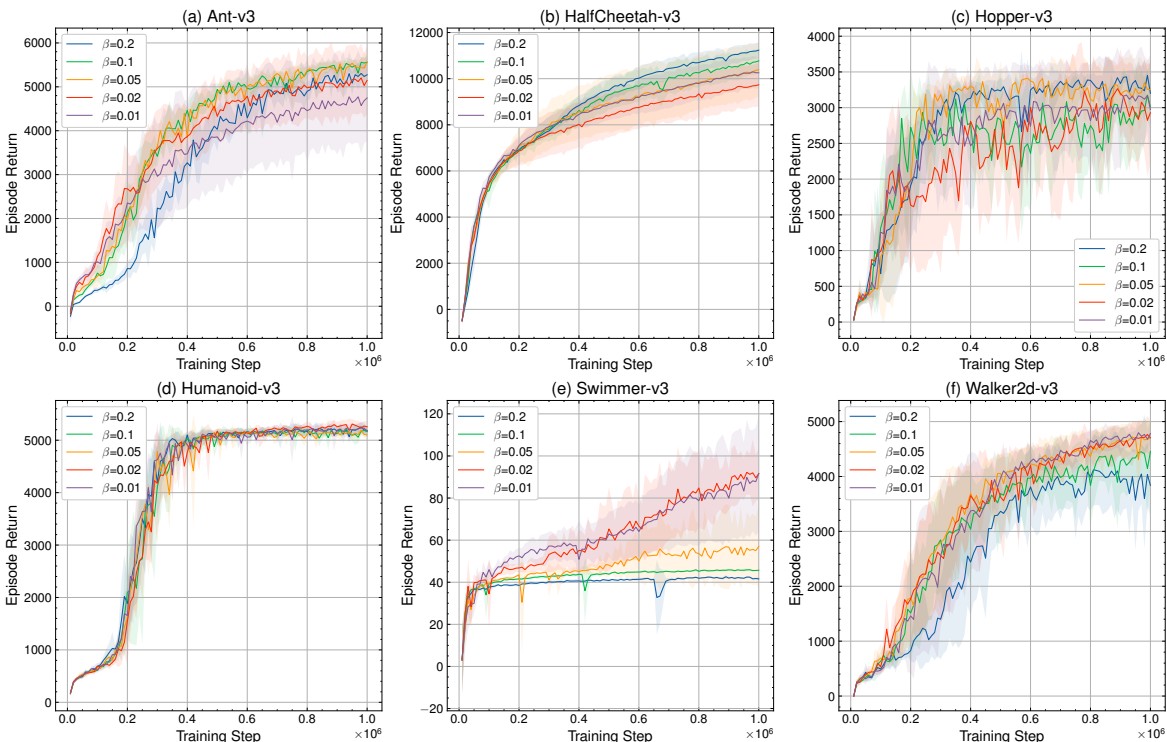

*Figure 7.* Learning curves of different temperature coefficients on Mujoco benchmarks.

## D.4. Testing the Accuracy of the Proposed Numerical Integration Technique on Probability Approximation

In theory, the numerical integration will be accurate when the diffusion step $T$ and the number of samples $N$ for probability approximation become large enough, according to the Law of Large Numbers. To exhibit the accuracy of different $T$ and $N$, we conduct experiments on a simple 2-D toy example where the target distribution $p(x)$ is a mixture of four Gaussian distributions with equal weights. Therefore, we set $Q(x) = \log p(x)$ and utilize the QNE method proposed in our paper to train a diffusion model. We display the approximation results of different $T$ and $N$ in Figure 11. The setting $T = 20, N = 50$ used in the paper can provide an effective probability approximation for the diffusion policy. When the samples are less ($T = 20, N = 20$), although there is a non-negligible error to the ground truth, the numerical integration method can still assign higher values for the region with higher probability. In this case, the estimated log probability can be considered as a kind of intrinsic curiosity reward to promote the exploration of the action region with low policy probability.

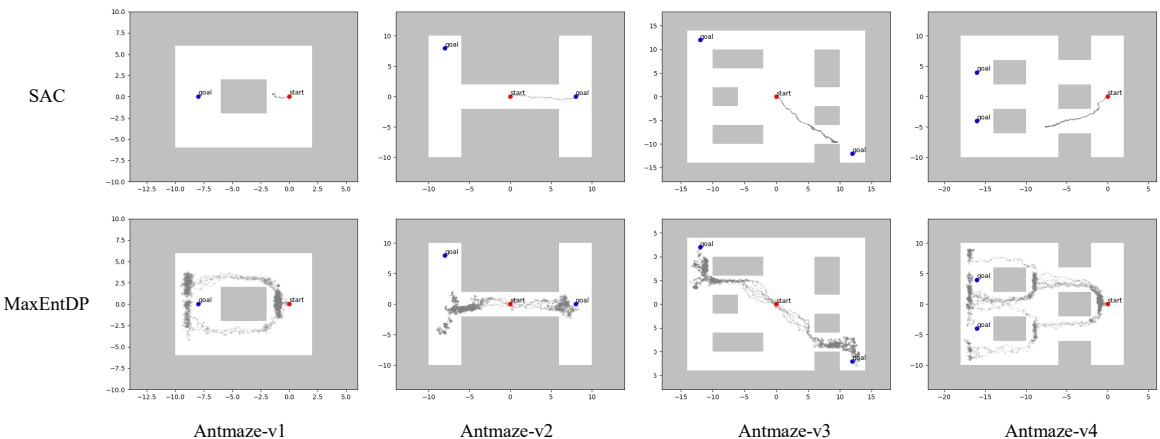

*Figure 8.* Trajecories generated by MaxEntDP and SAC after 1M environment interactions in Antmaze benchmarks.

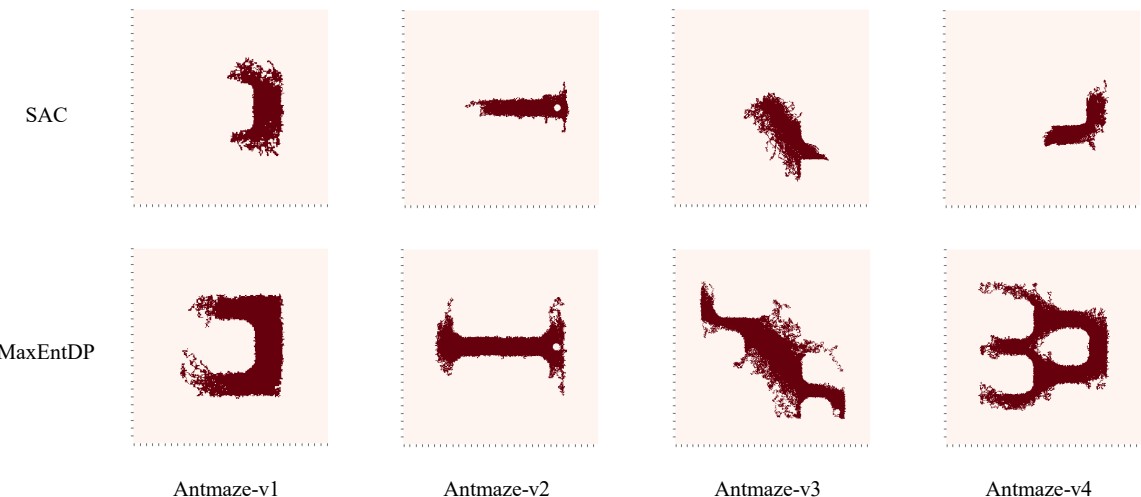

*Figure 9.* State coverage of MaxEntDP and SAC after 1M environment interactions in Antmaze benchmarks. MaxEntDP can explore different behavior modes at the same time and show broader state coverage than SAC, exhibiting efficient exploration of the high-dimensional state-action space.

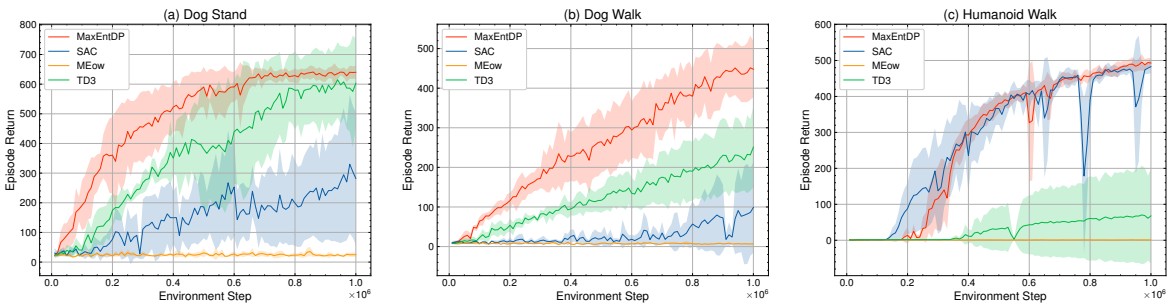

*Figure 10.* Learning curves on DeepMind Control suite. The solid lines are the means, and the shaded regions represent the standard deviations over five runs.

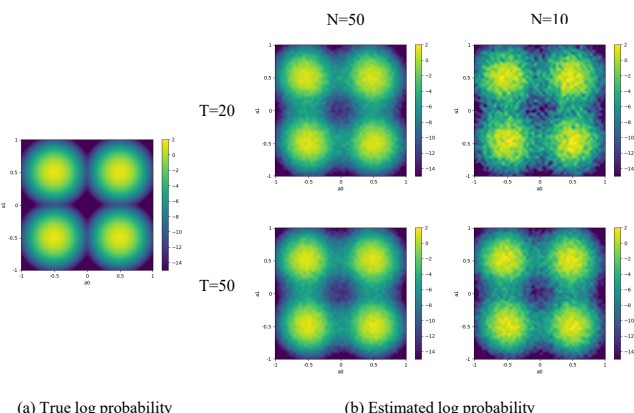

(a) True log probability          (b) Estimated log probability

*Figure 11.* The probability approximation using numerical integration method on a 2-D toy example with different diffusion steps $T$ and sample numbers $N$. The target distribution is a mixture of four Gaussian distributions, whose means are (-0.5, -0.5), (-0.5, 0.5), (0.5, 0.5) and (0.5, -0.5). The standard deviations and weights of the four components are the same, which are 0.1 and 0.25, respectively. The setting in the paper ($T = 20, N = 50$) can provide an effective approximation for the true log probability. When fewer samples ($T = 20, N = 10$) are used, despite some estimation errors, our method can still assign higher values to high-probability regions.

