# OpenReview forum: "Maximum Entropy Reinforcement Learning with Diffusion Policy"
_ICML.cc/2025/Conference — ICML 2025 poster_

### Official Review · Reviewer_TSPJ · 2025-03-11

**Overall Recommendation:** 3

**Summary:**

This paper focuses on adapting diffusion-based policies to maximum entropy reinforcement learning (MaxEnt RL) for better exploration. The primary obstacles are: 1) policy evaluation involves computing the log-probability over the clean actions, which for diffusion policies is non-trivial; and 2) policy improvement requires aligning the diffusion with the Boltzmann distribution of the Q-value functions, which we cannot readily access. To tackle these challenges, this paper proposes Q-weighted Noise Estimation and borrowed the probability calculation method from previous literature, which leads to a coherent algorithm that accommodates diffusion-based policies while also strictly corresponding to MaxEnt RL in theory. The evaluation is conducted on classical MuJoCo locomotion tasks, and the proposed method, MaxEntDP demonstrates improvements over both diffusion-based and non-diffusion algorithms.

## update after rebuttal
The authors demonstrated in their last response that MaxEntDP achieves significantly better performance over SAC on dog tasks from DMControl, which validates the benefit of combining diffusion policies with the MaxEnt RL framework. In light of this, I decided to raise my score, and I encourage the authors to conduct experiments on the DMControl benchmark with other baseline algorithms as well.

However, some theoretical concerns still remain. As I mentioned in my initial review, the assumption that $\epsilon_\phi$ is a well-trained noise prediction network is too strong since the policy improvement and the policy evaluation steps are interleaved. This compromises the accuracy of the estimated log-probability and may cause potential instabilities during training.

Therefore, given these observations and discussions, I will keep my evaluation as Weak Accept.

**Claims And Evidence:**

Yes, most of the claims in the paper are supported by theory or empirical evidence.

**Essential References Not Discussed:**

The idea of sampling from an energy-based distribution, of which we only have the potential function but no samples, is also related to model-based diffusion [1].

[1] Pan, Chaoyi, et al. "Model-based diffusion for trajectory optimization." Advances in Neural Information Processing Systems 37 (2024): 57914-57943.

**Experimental Designs Or Analyses:**

Yes, the experiments validate the main claims in this paper, including the superior performance of MaxEntDP, the variance reduction property of QNE, and the necessity of entropy in policy evaluation.

However, from current results it seems that the benefits of MaxEntDP is marginal, and in fact some baseline methods like SAC may achieve better performances if allowed to tune its hyper-parameters. I suggest to include experiments on harder online RL benchmarks, such as DMControl, and keeping the hyper-parameter tuning effort the same across baseline algorithms, to further validate the benefits such as improved exploration.

**Methods And Evaluation Criteria:**

MaxEntDP inherits the popularized MaxEnt RL framework, which is known to be effective for online RL due to enhanced exploration brought by entropy regularization. As for the benchmarks, although Gym-MuJoCo is the most commonly used benchmark in online RL, it is comparatively simple due to dense rewards and lower DoF.  I encourage the authors to also include results on tasks like MetaWorld or DMControl, where the task is more complex and the necessity of exploration is more pronounced.

**Other Comments Or Suggestions:**

My main concerns are discussed in "Theoretical Claims" and "Experimental Designs Or Analyses", please refer to these sections.

**Other Strengths And Weaknesses:**

The development of the problem, the motivation, and the method are clear and easy to follow.

**Questions For Authors:**

N/A

**Relation To Broader Scientific Literature:**

1. Using importance sampling to sample from the posterior distribution is similar, although not the same, to the technique used in iDEM [1].
2. The analytical computation of the log-probability is inspired by ITDiffusion [2]. The authors changed the integration variable for numerical stability.

[1] Akhound-Sadegh, Tara, et al. "Iterated denoising energy matching for sampling from boltzmann densities." arXiv preprint arXiv:2402.06121 (2024).

[2] Kong, Xianghao, Rob Brekelmans, and Greg Ver Steeg. "Information-theoretic diffusion." arXiv preprint arXiv:2302.03792 (2023).

**Theoretical Claims:**

First of all, the theory largely builds upon the analysis from MaxEnt RL. For the policy improvement step, this paper adopts an importance-weighted sampling technique similar to iDEM to reversely sample from the posterior $p(a_0|a_t)$. To reduce the variance of importance sampling, importance re-sampling is used to derive a biased objective but with reduced variance. The variance reduction property has been demonstrated by previous literature.

For the policy evaluation step, the analytical calculation for log probability is also borrowed from previous literature. However, Corollary 3.4 holds under the assumption that $\epsilon_\phi$ is a well-trained noise prediction network. During RL training, **such an assumption may not hold** since the policy improvement and evaluation are interleaved and we cannot guarantee that the policy network is well-trained. Under this circumstance, Eq. 20 does not exactly correspond to the log-probability of the actual action distribution.

---

> ### Author Rebuttal · Authors · 2025-04-01
>
> Thank you for your valuable feedback and suggestions. Here, we aim to address the questions raised in the review.
>
> >**Q1: Include results on tasks like MetaWorld or DMControl.**
>
> We compare MaxEntDP with SAC on DMControl and MyoSuite benchmarks. The results are shown in https://anonymous.4open.science/api/repo/pics-E459/file/DMC_myo.pdf?v=538c572b.
>
> >**Q2: The importance-weighted sampling technique in the paper is similar to iDEM.**
>
> We would like to emphasize that although the expression of our QNE method appears similar to iDEM [1], our method exhibits significantly lower estimation variance and steady performance improvement throughout the training process over the same method replaced with iDEM (Figure 3). In addition, our method does not require gradient computation
> of the Q-function as in iDEM and thus is more computationally
> efficient. These advantages demonstrate the superiority of our QNE method on diffusion policy optimization.
>
> >**Q3: Concern on the assumption of Corollary 3.4.**
>
> We use the target Q network to train the diffusion policy. Since we adopt the Exponential Moving Average (EMA) update to smooth the change of the target Q network, it is not difficult to learn a well-trained policy network.
>
> >**Q4: Keep the hyper-parameter tuning effort the same across baseline algorithms.**
>
> For fair comparison, we unify shared hyperparameters (batch size, discount factor, the depth and width of hidden layers, learning rate and replay buffer size) across baseline algorithms. Other hyperparameters strictly follow the settings specified in the original paper or codebase, which are already tuned by their authors.
>
> >**Q5: One related work model-based diffusion [2] is missing.**
>
> We will cite it in revised versions of the paper. The model-based diffusion proposes the Monte Carlo estimation for computing the score function and uses the Monte Carlo score ascent to generate samples following the Boltzmann distribution of a given function. This method is similar to our QNE method, however, QNE has several properties which matter in RL training:
> 1. We use a parameterized network to approximate the scaled score function, while the model-based diffusion needs to compute the score function using Monte Carlo estimation when generating samples. Therefore, sample generation of diffusion-based diffusion is time-consuming, which will slow the training speed of the RL algorithms.
> 2. We adopt the ancestral sampling in DDPM to generate samples, that are more diverse than that of Monte Carlo score ascent used in model-based diffusion.
> 3. We propose to modify the standard Gaussian to the truncated Gaussian in QNE to model the action distribution with a bounded action space. However, model-based diffusion can not address such a bounded distribution.

---

> > ### Comment · Reviewer_TSPJ · 2025-04-03
> >
> > I appreciate the author's response. However, I still have some remaining concerns:
> >
> > 1. It's still hard to interpret the benefit of the diffusion policy from the results on DMControl and MyoSuite. Why don't you include results on harder environments, such as humanoid tasks or dog tasks from DMControl?
> >
> > 2. The assumption of a fully converged policy network at every iteration still appears overly restrictive to me, although it seems that its effect on performance is negligible (Figure 6.b). Besides, what is the cost of computing log-probability? Is it computationally heavy?
> >
> > 3. For Q4, I was actually referring to the temperature coefficient $\beta$. Is it possible to auto-tune this hyperparameter like SAC?

---

> > > ### Author Response · Authors · 2025-04-09
> > >
> > > Thanks for your constructive feedback. Below we would like to address your remaining concerns.
> > >
> > > >**Q1: Add experiments on harder environments, such as humanoid tasks or dog tasks from DMControl.**
> > >
> > > We compare MaxEntDP with SAC on humanoid and dog benchmarks from DMControl, showing the results in https://anonymous.4open.science/api/repo/pics-E459/file/DMC_plus.pdf?v=9a37bdf8. Our MaxEntDP outperforms SAC on these challenging high-dimensional RL tasks.
> > >
> > > >**Q2: The cost of computing log-probability.**
> > >
> > > In the following table, we list the training time on HalfCheetah-v3 benchmark with different $N$, which is the sample number per timestep for probability estimation. Since the samples can be processed parallelly, probability computation does not cause a heavy burden in our experiments.
> > >
> > > | $N$ | 0   | 10   | 20   | 50   | 100   |
> > > | ----------------- | ------ | ------ | ------ | ------ | ------ |
> > > | Training time (h)      | 2.3  | 2.7  | 3.4  | 3.9  | 5.8  |
> > >
> > >
> > > >**Q3(a): Tune temperature coefficients for SAC.**
> > >
> > > We initialize SAC with the same temperature coefficient as MaxEntDP and keep it fixed during the training process. The corresponding results are displayed in https://anonymous.4open.science/api/repo/pics-E459/file/comparison_fixed_temp.pdf?v=a29784b9. MaxEntDP still outperforms the SAC variant with fixed temperature coefficients.
> > >
> > > >**Q3(b): Is it possible to auto-tune temperature coefficients like SAC?**
> > >
> > > SAC sets a target entropy ($-|\mathcal{A}|$) and updates the temperature coefficient based on the distance between the current entropy and its target. Since MaxEntDP also computes the log probability in the policy evaluation step, it is feasible to apply the same method to auto-tune the temperature coefficient. However, in our experiments, we cannot find a unified target entropy that can perform well in all environments. That may be because diffusion policy is much more complex than Gaussian policy, making the simple linear function unable to express the relation between the best target entropy and the dimension of the action space. We leave exploring suitable methods to auto-tune temperature coefficients for MaxEntDP to future work.

---

### Official Review · Reviewer_7ggH · 2025-03-11

**Overall Recommendation:** 3

**Summary:**

This paper introduces Maximum entropy Reinforcement Learning with Diffusion Policy (MaxEntDP). More specifically, this method proposes solutions to the well-known problems on how to approximate the target distribution composed of the exponential of the Q-function and how to calculate the log likelihoods of the marginal distribution of diffusion models. Both problems are essential to train in the maximum entropy RL framework and need careful treatment when using diffusion models.

## Updated after rebuttal
I adjusted my score after the authors have clarified my questions and concerns

**Claims And Evidence:**

This work proposes a solution on approximating the likelihood of the marginal distribution of diffusion models and proposes an objective to train the policy's score function without evaluating the gradient of the Q-function.

The claims are backed up with mathematical derivations and proofs and ablation studies.

**Essential References Not Discussed:**

To my knowledge, all essential references are discussed w.r.t. diffusion models. However, the paper lacks recent works in the field of maximum entropy reinforcement learning such as Bigger Regularized optimistic (BRO) [1] and CrossQ [2].

[1] Nauman, Michal, et al. "Bigger, Regularized, Optimistic: scaling for compute and sample efficient continuous control." The Thirty-eighth Annual Conference on Neural Information Processing Systems.

[2] Bhatt, A., et al. "CrossQ: Batch Normalization in Deep Reinforcement Learning for Greater Sample Efficiency and Simplicity." International Conference on Learning Representations (ICLR). 2024.

**Experimental Designs Or Analyses:**

From my understanding there are no experimental environments designed specifically for this work.

**Methods And Evaluation Criteria:**

The considered benchmark makes sense, though there are more sophisticated tasks in RL that should also be considered, e.g., high-dimensional control tasks from the DeepMind control suite are harder learning tasks that usually have a longer horizon than the mujoco environments from gym.

**Other Comments Or Suggestions:**

please see my comments from before.

**Other Strengths And Weaknesses:**

In general, the paper is well written and the reader can follow it. It is well-motivated and considers an important aspect of research for using diffusion models in reinforcement learning.
However, there are several points that need clarification:

- Lines 220 following mention that the distribution for the noisy actions p(a_t) is unknown but can be substituted with other distributions with full support. What does this mean? How is this justified given that the expectation is not correctly evaluated? Covering full support can be problematic for high-dimensional action spaces. This seems to be a major issue but was not discussed. Additionally, the considered environments do not consider high-dimensional action spaces, so it is hard to judge whether this is a bottleneck.

- The paper states that a truncated standard Gaussian is used to keep the sampled actions within the bounds. This, however, changes the likelihood to my understanding, but no correction is mentioned. For example, SAC applies the change of variables and corrects the likelihood in this way. This is however not considered here.

- While there are some tasks where the proposed method performs better compared to Gaussian-based policies, it does not perform better than other Diffusion-Based methods.

- More experiments on more sophisticated tasks with longer horizons and higher dimensional action spaces, such as those from the Deepmind control suite, myo suite, or the humanoid bench, would strengthen the paper.

- The learning curves plot the "training step" but it is never mentioned what this means. I am confused as RL papers usually plot the number of environment interactions instead of some other metric. This needs clarification.

- Minor note: K-L instead of KL in line 119 right column

**Questions For Authors:**

please see my comments from before.


------ I adjusted my score after the authors have clarified my questions and concerns ------

**Relation To Broader Scientific Literature:**

The work relates well to prior findings and distinguishes itself from prior works.

**Theoretical Claims:**

I did not check the proofs for Theorem 2.1

I checked the proofs to Theorem 3.1, Theorem 3.2, Theorem 3.3, which seem to be fine to me.


I also checked the proof for the iDEM derivation (A4) and have a concern:
Eq 59 pulls the gradient w.r.t. a_0 in front of w(a_0), but this is mathematically not correct as w(a_0) is depending on a_0 as well. How is this justified?


I also checked the proof for Theorem 3.5. but have difficulties understanding it: How is the replacement of the integration domain in Eq. 65 justified? I don't understand this step. Intuitively, it is a very strong change integrating from 0 to 1 instead of from -infinity to infinity. It is difficult to judge whether the approximation of calculating the log likelihood is a good approximation. My concern is also strengthened by the fact that the learning curves with the entropy bonus are actually not significantly better than the learning curves without the entropy (Fig. 5 in Section 5).

---

> ### Author Rebuttal · Authors · 2025-04-01
>
> Thank you for your thoughtful review and constructive feedback. Below, we will address your comments and hope that this clarifies the context of our work.
>
> >**Q1: Consider more sophisticated RL tasks.**
>
> We compare MaxEntDP with SAC on DMContorl and MyoSuite benchmarks. The results are shown in https://anonymous.4open.science/api/repo/pics-E459/file/DMC_myo.pdf?v=538c572b.
>
> >**Q2: Justification of Eq 59.**
>
> We are sorry for omitting the parentheses on $\bigtriangledown_{a_0} w(a_0)$ in Eq 59, which causes trouble in understanding. In fact, we apply the integration by parts formula (line 777) to derive Eq 59, i.e., $\int_{\Omega} v \nabla u \, d\Omega = \int_{\Gamma} u v n \, d\Gamma - \int_{\Omega} u \nabla v \, d\Omega$ (consider $w(a_0)$ as $v$ and $N(a_0)$ as $u$; the term $\int_{\Gamma} u v n \, d\Gamma$ is zero since $w(a_0)$ and $N(a_0)$ decay rapidly at infinity).
>
> >**Q3: Justification of Eq 65.**
>
> In Eq 65, we change the integration variable from $\alpha_t$ to $\sigma(\alpha_t)$ using the equation $\alpha_t=\log \frac{\sigma(\alpha_t)}{1-\sigma(\alpha_t)}$, as illustrated in line 807. Then the integration domain of Eq 65 has been changed to the varying range of the new integration variable $\sigma(\alpha_t)$, which is $(0,1)$.
>
> >**Q4: The accuracy of the log-likelihood approximation.**
>
> Due to the length limit of rebuttal, please refer to our response to the Q2 of reviewer fZmT.
>
> >**Q5: The paper lacks recent work such as BRO and CrossQ.**
>
> We will cite them in future versions. Briefly, the two methods propose some improvements to the SAC algorithm. BRO develops the advanced BroNet architecture, regularization, and optimistic upper-bound Q-value approximation. CrossQ removes the target networks and adopts Batch Normalization for high sample efficiency. Since the improvements of the two methods are also compatible with our MaxEntDP algorithm, it is interesting to see how the performance of MaxEntDP can be enhanced by combining these improvements.
>
> >**Q6: Why can the noisy actions** $p(a_t)$ **be substituted with other distributions?**
>
> As shown in Eq17, the training target $\epsilon^{*}$ of the noise prediction network depends only on its input $(a_t,\alpha_t)$, which means that we can minimize the L2 loss for each input point. Thus, the noisy action samples from other distributions can also be used for training. This can be seen as a kind of "off-policy" training. Moreover, in the paper we diffuse the action samples in the replay buffer to obtain the surrogate distribution of $p(a_t)$ (line 295). As the training progresses, the actions generated by the diffusion policy will be closer to the target distribution $p(a_0)$, then the diffused distribution of these actions will also be closer to the true distribution of $p(a_t)$, consequently, the training will be more and more "on-policy".
>
> >**Q7: The reason for using a truncated standard Gaussian in RL tasks with bounded action spaces.**
>
> We would like to point that MaxEntDP and SAC model the bouded policy distribution in different ways. SAC outputs an unbounded distribution and transforms it to a bounded one by applying a tanh function, while MaxEntDP directly models a bounded distribution following the common practices in the image generation domain. To learn this bounded distribution, we need to change the target distribution to $p(a_0)\propto\exp(\frac{1}{\beta}Q(a_0))I_{a_0\in[lb,ub]}$, where $I_{a_0\in [lb,ub]}$ is an indicator function to check if $a_0$ lies within the bound. Similar to lemma 3.1, we can obtain that the reverse transition distribution has be changed to $p(a_0|a_t)\propto\exp(\frac{1}{\beta}Q(a_0)) I_{a_0\in[lb,ub]}
>      \mathcal{N}(a_0|\frac{1}{\sqrt{\sigma(\alpha_t)}} a_t,\frac{\sigma(-\alpha_t)}{\sigma(\alpha_t)}I)$. Taking the indicator function and the Gaussian distribution together, we can derive a truncated Gaussian distribution with a bound of $[lb,ub]$. Then, the conditional distribution $p(a_0|a_t)$ can be seen as a truncated Gaussian distribution of $a_0$ weighted by the exponential of the Q-function. Therefore, for diffusion policy training with a bounded action space, we only need to modify the standard Gaussian distribution in the QNE method to a truncated Gaussian distribution.
>
> >**Q8: Performance comparison with other Diffusion-Based methods.**
>
> We would like to emphasize that none of the competing diffusion-based methods performs consistently well on all tasks (DACER struggles on Walker2d, and others underperform on HalfCheetah). However, our MaxEntDP displays consistent sample efficiency and stability across all tasks.
>
> >**Q9: The meaning of "training step" in the learning curves; K-L in line 119.**
>
> We use "training step" because both DACER and QVPO use it as the x-axis of the learning curve, which means the training steps of the actor/critic network. Since we use a UTD of 1, it is the same as the number of environment interactions. And we will modify the 'K-L' in line 119 in revised versions.

---

> > ### Comment · Reviewer_7ggH · 2025-04-04
> >
> > Thank you for your responses, which have clarified most concerns. However, I still have some questions.
> >
> > - Q6: I see the authors' point that 'other samples can be used for training'. The paper states that 'the true distribution of noisy actions ... may be inaccessible, we can substitute it with other distributions with full support'. My question is more related to the fact that another distribution with full support is used. How is it guaranteed that this distribution has full support?  Given that an approximation to p(a_t) is used, shouldn't Eq. 18 involve some techniques to correct the expectation? Or did I misunderstand something in this case?
> >
> > I appreciate the authors' efforts in running more experiments on the DMC suite. It seems that there is no big benefit against a Gaussian policy in this case. I would assume this is due to the rather low-dimensional tasks. Did the authors also analyze higher-dimensional tasks like the dog? Intuitively, a diffusion policy might show benefits for higher-dimensional learning tasks both in the observation and action dimensions.

---

> > > ### Author Response · Authors · 2025-04-09
> > >
> > > Thanks for your valuable feedback. Here we would like to answer your remaining questions.
> > >
> > > >**Q1: How is it guaranteed that** $p(a_t)$ **has full support?**
> > >
> > > In MaxEntDP, We use $a_t=\sqrt{\sigma(\alpha_t)}a_0+\sqrt{\sigma(\alpha_t)}\epsilon_t$ to obtain noisy action samples (line 295), where $a_0$ is an action sampled from the replay buffer and $\epsilon_t \sim \mathcal{N}(0,I)$. Since the distribution of $\epsilon_t$ has a full support over $\mathbb{R}^{|\mathcal{A}|}$, the distribution of $a_t$ also has a full support over $\mathbb{R}^{|\mathcal{A}|}$, i.e., we can sample any $a_t \in \mathbb{R}^{|\mathcal{A}|}$ at a non-zero probability.
> > >
> > > >**Q2: Given that an approximation to** $p(a_t)$ **is used, should Eq. 18 involve some techniques to correct the expectation?**
> > >
> > > When the network capacity is sufficient, whatever distribution of $p(a_t)$ with full support is used, the minimizer of Eq. 18 is the same, which is  $\epsilon_{\phi}(a_t,\alpha_t)=\epsilon^*(a_t,\alpha_t)$ for all $(a_t,\alpha_t)$. Therefore, in this case, we can use another distribution of $p(a_t)$ with full support without correction. When the network capacity is insufficient, changing $p(a_t)$ also changes the minimizer of Eq. 18, therefore, just as you said, correction is needed to assign proper weights to each $(a_t,\alpha_t)$. However, because the true distribution of $p(a_t)$ is unknown, applying a correction to the weights of $a_t$ is intractable. In addition, in our experiments, we do not find that the lack of such correction causes trouble in diffusion policy optimization.
> > >
> > > >**Q3: Add experiments on higher-dimensional tasks.**
> > >
> > > We compare MaxEntDP with SAC on humanoid and dog benchmarks from DMControl, showing the results in https://anonymous.4open.science/api/repo/pics-E459/file/DMC_plus.pdf?v=9a37bdf8. Our MaxEntDP outperforms SAC on these challenging high-dimensional RL tasks.

---

### Official Review · Reviewer_fZmT · 2025-03-13

**Overall Recommendation:** 3

**Summary:**

This paper introduces MaxEntDP, a new online diffusion-based RL algorithm that integrates diffusion models into the maximum entropy framework. The method proposes a Q-weighted noise estimation for policy improvement and use numerical integration to estimate action probability for policy evaluation. Experiments are conducted on MuJoCo environment, comparing with classic MaxEnt algorithms and other online diffusion-based algorithms, verifying the effectiveness of the proposed method.

**Claims And Evidence:**

Overall, the authors make several claims but only partially supported. First, in Section 3, the authors claim that MaxEnt RL with expressive diffusion model can capture multimodal behaviors. This is evidenced by the multi-goal 2D toy example. However, I am suspicious how this will generalize to high-dimensional tasks because the goals in the toy example is easy to explore. For example, in DDiffPG [1] it provides several high-dim multi-goal tasks, and it is interesting to see if MaxEnt RL can learn different solutions too. Second, the authors claim that by maximizing the entropy MaxEnt RL improves the exploration. However, there is no experiment or analysis to support it. It would be great to include some visualizations or state coverage in the experiment. Moreover, the authors say numerical integration provides an effective approximation and MaxEntDP achieves the optimal MaxEnt policy given sufficient model capacity. Both are not supported without any evidence or comparison.

[1] Li, Steven, et al. "Learning multimodal behaviors from scratch with diffusion policy gradient." Advances in Neural Information Processing Systems 37 (2024): 38456-38479.

**Essential References Not Discussed:**

One key claim in Section 3 and Section 5.1 is to learn a multi-modal policy, however one related work DDiffPG is missing [1].
When comparing to learning curve, the current SOTA BRO is missing [2].

**Experimental Designs Or Analyses:**

The experiments confirm the effectiveness of the proposed method. However, it does not include a comparison with the current state-of-the-art BRO [2]. Additionally, its performance shows only a marginal improvement over existing diffusion-based baselines.

[2] Nauman, Michal, et al. "Bigger, regularized, optimistic: scaling for compute and sample-efficient continuous control." arXiv preprint arXiv:2405.16158 (2024).

**Methods And Evaluation Criteria:**

The paper proposes a novel Q-weighted noise estimation for diffusion policy optimization and an approximation for the diffusion model action probability. One problem is that both parts require monte-carlo sampling, which possess a challenge on estimation accuracy. In appendix the authors provide ablations on the key hyper-paramaters, in which the high variance verifies the instability. Moreover, the paper provides two tricks to stabilize and improve the performance. The action selection for inference is a common approach in offline RL but unusual in online setup. However, most of baselines do not use this action selection for a fair comparison as shown in appendix.

The method is evaluated only on MuJoCo tasks, which has almost no exploration challenge. I would suggest to experiment on more challenging environments, e.g., myosuite to showcase the exploration benefit from the MaxEnt framework.

**Other Comments Or Suggestions:**

N/A

**Other Strengths And Weaknesses:**

The paper is well-written and easy-to-follow.

**Questions For Authors:**

See above.

**Relation To Broader Scientific Literature:**

The paper is highly relevant to RL and diffusion model.

**Theoretical Claims:**

I have checked the proofs in the paper.

---

> ### Author Rebuttal · Authors · 2025-04-01
>
> Thank you for your valuable feedback and suggestions. Below we will address each concern raised in the review.
> >**Q1: Can MaxEntDP learn different solutions in high-dimensional multi-goal tasks from DDiffPG? And provide evidence for improved exploration.**
>
> We tested MaxEntDP and SAC on four versions of the AntMaze environment in DDiffPG (using dense rewards) and visualized the generated trajectories in Figure 1 of https://anonymous.4open.science/api/repo/pics-E459/file/antmaze.pdf?v=eebbbe1f. The results confirm that MaxEntDP can learn diverse behavior modes even in challenging high-dimensional RL tasks, while SAC fails to learn different solutions.
> In addition, we visualized state coverage for MaxEntDP and SAC (see Figure 2 in the above link). The results show that MaxEntDP explores multiple behavior modes and exhibits broader state coverage. This highlights the advantage of using a diffusion policy for efficient exploration.
>
> >**Q2: No evidence supporting numerical integration as an effective approximation.**
>
> According to the Law of Large Numbers, numerical integration accuracy improves with larger diffusion steps $T$ and sample numbers $N$. To exhibit the accuracy of different $T$ and $N$, we conducted experiments on a 2D toy example (a mixture of four Gaussians) and presented the results in https://anonymous.4open.science/api/repo/pics-E459/file/probability.pdf?v=86f81805. As shown in the figure, our setting in the paper ($T=20, N=50$) provides an effective probability approximation. Moreover, when fewer samples ($T=20, N=10$) are used, despite some estimation errors, our method still assigns higher values to high-probability regions, which can be considered as an intrinsic curiosity reward to promote exploration on the action region with low policy probability.
>
> >**Q3: The claim that MaxEntDP achieves the optimal MaxEnt policy with sufficient model capacity.**
>
> We do not make such a claim in the paper. Instead, we argue that incorporating a diffusion policy into MaxEnt RL improves exploration and moves the policy closer to the optimal solution, as proposed in the abstract. The 2D toy example in the paper reveals that MaxEntDP can explore the state-action space efficiently and finally learn a multimodal policy. The Mujoco experiments show performance improvements over other generative models and diffusion-based approaches, supporting our main claim.
>
> >**Q4: Concern about Monte Carlo sampling.**
>
> Your concern is reasonable since a higher estimation accuracy requires more Monte Carlo samples. However, in our experiments, we find that a small sample number of 1000 (for both diffusion policy optimization and action probability estimation) is enough to obtain a good performance. Additionally, as the samples can be processed parallelly and GPU throughput continues to improve, Monte Carlo sampling is not a bottleneck.
>
> >**Q5: Necessity of action selection and its application to all baselines.**
>
> Action selection is crucial because our method approximates the exponential of the Q-function, generating both high- and low-return actions. While beneficial for exploration, this can reduce test-time performance. To mitigate this, we apply action selection (only in testing) to pick the action with the highest Q-value. Similar techniques are used in SAC (take the Gaussian mean) and EBFlow. To ensure fairness, we applied action selection to all diffusion-based baselines (with a candidate number of 10) and reported results in https://anonymous.4open.science/api/repo/pics-E459/file/comparison_diffusion_selection.pdf?v=a952c896. Our MaxEntDP continued to demonstrate high sample efficiency and stability across all tasks.
>
> >**Q6: Add experiments in more challenging environments.**
>
> We compared MaxEntDP with SAC on the DMControl and MyoSuite benchmarks. Results are shown in https://anonymous.4open.science/api/repo/pics-E459/file/DMC_myo.pdf?v=538c572b.
>
> >**Q7: Include a comparison with BRO.**
>
> BRO improves SAC with advanced network architectures, regularization, and optimistic Q-value estimation. Since BRO's enhancements can be applied to any baseline (non-diffusion and diffusion-based) and MaxEntDP, a direct comparison would be unfair. However, integrating BRO's improvements into MaxEntDP is an interesting future direction.
>
> >**Q8: Performance gains of MaxEntDP over existing diffusion-based baselines.**
>
> None of the competing diffusion-based methods consistently outperforms across all tasks—e.g., DACER struggles on Walker2d, and others underperform on HalfCheetah. In contrast, MaxEntDP exhibits consistent sample efficiency and stability across all tasks.
>
> >**Q9: Missing reference to DDiffPG.**
>
> Thanks for your valuable suggestion. We will cite it in future revisions. While DDiffPG explicitly distinguishes different behavior modes and learns a Q-function for each mode, MaxEntDP employs a single Q-function, making it simpler and more computationally efficient.

---

### Decision · Program_Chairs · 2025-05-01

**Decision:**

Accept (poster)

**Comment:**

**Summary:**
This paper proposes MaxEntDP, a diffusion-based reinforcement learning framework that integrates the maximum entropy principle into the training of expressive diffusion policies. The method addresses key challenges in policy evaluation and improvement via Q-weighted noise estimation and numerical approximation of marginal likelihoods, showing promising empirical results and solid theoretical foundations, although some concerns about scalability, practical assumptions, and comparisons remain.

**Pros:**
1. Presents a novel integration of diffusion models into the MaxEnt RL framework, enabling more expressive and exploratory policy learning.
2. Provides sound theoretical analysis and well-motivated algorithmic components, supported by ablations and baseline comparisons.
3. Demonstrates performance gains on standard MuJoCo benchmarks, with indications of promise in more complex settings.

**Cons:**
1. Key assumptions (e.g., well-trained noise networks and full support approximations) may not fully hold during interleaved training phases.
2. Experimental evaluations are limited to relatively simple environments, with marginal gains over existing methods and missing recent SOTA baselines.
3. Some theoretical derivations and implementation choices (e.g., integration bounds, likelihood estimation) require further justification and clarity.